# KAST-BAR: Knowledge-Anchored Semantically-Dynamic Topology Brain Autoregressive Modeling for Universal Neural Interpretation

Haoning Wang [1]   Wenchao Yang [1]   Shuai Shen [1]   Yang Li [1 2 3 4]

## Abstract

While EEG foundation models have shown significant potential in universal neural decoding across tasks, their advancement remains constrained by the inadequacy modeling of *complex spatiotemporal topology*, as well as the inherent *modality gap* between low-level physiological signals and high-level textual semantics. To address these challenges, we propose a **K**nowledge-**A**nchored **S**emantically-Dynamic **T**opology **B**rain **A**uto**r**egressive Model (KAST-BAR), which dynamically aligns physiological representations derived from multi-level brain topology with an expert-level semantic space. Specifically, we design a Dual-Stream Hierarchical Attention (DSHA) encoder that accurately captures the brain's intrinsic non-Euclidean topology by modeling local temporal dynamics with global spatial contexts. On this basis, a Knowledge-Anchored Semantic Profiler (KASP) is proposed to synthesize physically-grounded and instance-level textual profiles, which subsequently drive a Semantic Text-Aware Refiner (STAR) to dynamically reconstruct EEG representations using Latent Expert Queries. By conducting large-scale pre-training on 21 diverse datasets to build a foundation model, KAST-BAR effectively integrates expert-level medical knowledge into EEG signal representations, consistently achieving superior performance across six downstream tasks. Our code is available at https://github.com/KAST-BAR/KAST-BAR

[1]School of Automation Science and Electrical Engineering, Beihang University, Beijing, China. [2]School of Biological Science and Medical Engineering, Beihang University, Beijing, China. [3]State Key Laboratory of Virtual Reality Technology and Systems, Beihang University, Beijing, China. [4]7T Magnetic Resonance Imaging Translational Medical Center, Department of Radiology, Southwest Hospital, Army Medical University (Third Military Medical University), Chongqing, China.. Correspondence to: Yang Li <liyang@buaa.edu.cn>.

*Proceedings of the 43rd International Conference on Machine Learning*, Seoul, South Korea. PMLR 306, 2026. Copyright 2026 by the author(s).

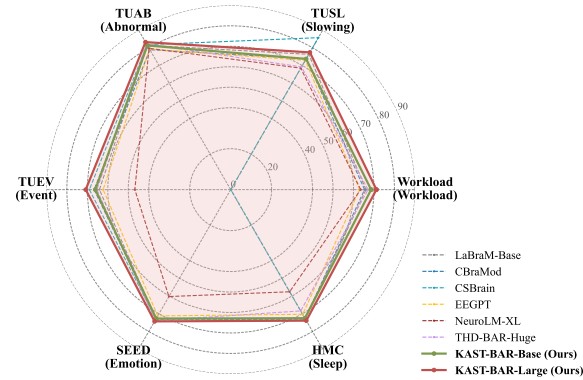

*Figure 1.* Holistic evaluation of KAST-BAR against leading methods across six diverse EEG datasets.

## 1. Introduction

The emergence of foundation models has reshaped the interpretation paradigm of physiological signals, particularly large-scale electroencephalograph (EEG) pre-trained models, which have demonstrated unprecedented potential in decoding brain neural activity by mining massive unlabeled EEG data (Jiang et al., 2024; 2025; Yang et al., 2025; Wei et al., 2025). These models strive to learn generalized representations transferable to diverse neurological tasks. However, despite recent significant progress, constructing a truly universal and interpretable EEG foundation model still faces the following key challenges:

**1) Insufficient modeling of Non-Euclidean Manifold and Spatiotemporal Topology.** The crux of existing bottlenecks lies primarily in the systematic deficiency in modeling the non-Euclidean spatiotemporal topology of EEG. EEG signals are intrinsically distributed on complex 3D manifold, whereas mainstream paradigms often enforce dimensionality reduction, severely disrupting the signal's inherent geometric structure(Yi et al., 2023). Although recent works (Chen et al., 2026; Yang et al., 2025) have attempted multiscale and hierarchical improvements, most methods still rely on implicit encoding (Wang et al., 2025a; Zhou et al., 2025; Jiang et al., 2025; Li et al., 2025b), failing to establish explicit information interaction between different topological levels. This insufficient modeling of microscopic features and macroscopic contexts leads to the irreversible loss of critical spatiotemporal evolutionary information.

**2) Cross-Modal Semantic Gap and Alignment Dilemma.** The natural modal barrier between the underlying physical attributes of EEG data and the high-level semantic representations of text data makes establishing a precise mapping between them extremely difficult. Directly utilizing general corpora lacking medical knowledge for geometric spatial alignment fails to construct a semantically precise, consistent, and interpretable representation space (Jiang et al., 2025). This dilemma forces existing alignment strategies into a trade-off: one category is forced to retreat to simplified general text instructions, leading to the loss of EEG physical attributes and task specificity (Wang et al., 2025b; Jiang et al., 2026); while the other relies heavily on scarce, unstructured clinical reports (Gijsen & Ritter, 2025). More critically, such alignment paradigms, lacking task-context interaction mechanisms, struggle to accommodate the heterogeneity of subject specificity and task distributions. Faced with EEG tasks exhibiting distinct spectral and topological patterns, existing "one-size-fits-all" strategies or simple Mixture-of-Experts (MoE) mechanisms lack task-aware adaptive capabilities (Li et al., 2025a; Wang et al., 2025a; Chen et al., 2026; Lu et al., 2025). This systematic absence of physical grounding, medical priors, and task adaptability ultimately hinders the establishment of precise, fine-grained signal-semantic mappings. **A detailed discussion of additional related works is provided in Appendix A**.

To address these challenges, we propose **KAST-BAR**, a novel foundation model that synergizes knowledge-driven semantic reasoning with dynamic topological perception for universal brain signal understanding and cross-modal analysis. Different from traditional implicit alignment strategies, KAST-BAR establishes an explicit "Topological Encoding - Knowledge Anchoring - Guided Refinement" paradigm: First, the model employs a Dual-Stream Hierarchical Attention (DSHA) encoder to perform topology-enhanced discretized encoding of EEG data on non-Euclidean 3D manifold by explicitly modeling local refinement with global spatiotemporal features; Subsequently, a Knowledge-Anchored Semantic Profiler (KASP) is introduced to synthesize expert-level medical insights highly correlated with the signals, facilitating the LLM's understanding of the task context; These insights further drive the Semantic Text-Aware Refiner (STAR) to generate latent expert queries, which dynamically aggregate and reconstruct EEG features. This architecture endows the LLM with the capability to jointly process prior knowledge, adaptive EEG summaries, and discrete embedding sequences throughout the entire pipeline of self-supervised encoding, autoregressive pre-training, and instruction fine-tuning, thereby acquiring cross-modal representations that are both robust and interpretable. Pre-trained on 21 datasets, KAST-BAR achieves superior performance across six tasks, validating its efficacy in infusing medi-

cal knowledge into cross-modal representations The main contributions of this paper are summarized as follows:

- We propose a Dual-Stream Hierarchical Attention encoder that explicitly models the brain's non-Euclidean 3D manifold through bidirectional information interaction, enabling structure-preserving discrete EEG tokenization.
- We pioneer a knowledge-driven semantic-topological orchestration mechanism, where the Knowledge-Anchored Semantic Profiler synthesizes medical priors to actively guide the Semantic Text-Aware Refiner, achieving granular neuro-semantic alignment.
- We propose KAST-BAR, an EEG foundation model pre-trained on a diverse corpus of 21 datasets. Demonstrating superior performance across six downstream tasks, our method validates the efficacy of infusing medical knowledge into biological signal modeling.

**Conflict of Interest Disclosure.** The authors declare that they have no financial conflicts of interest related to this work.

## 2. Methodology

In this section, we elaborate on the architectural design of our KAST-BAR. An overview of the KAST-BAR is shown in Figure 2, we first introduce the DSHA module for topology-enhanced discretized encoding of EEG features in Sec. 2.1; subsequently, we delve into the KASP-STAR synergy in Sec. 2.2 and Sec. 2.3, detailing how it achieves fine-grained cross-modal alignment via generated medical insights and dynamic queries.

### 2.1. Dual-Stream Hierarchical Attention Encoder

EEG signals differ fundamentally from image or text data regarding their spatial relationships; they are recorded from a three-dimensional non-Euclidean manifold, where the spatial arrangement of electrodes encodes rich functional connectivity relationships rather than simple 2D or 1D structures. To explicitly model these characteristics, we define the input EEG data as $X \in \mathbb{R}^{C \times T}$, with $C$ channels and $T$ time points. We first apply a temporal slicing operation, partitioning $X$ into non-overlapping segments of length $W$ to generate a patched sequence $X_{EEG} \in \mathbb{R}^{C \times P \times W}$, where $P = \lfloor T/W \rfloor$. These patches are encoded into shallow features $F_{EEG} \in \mathbb{R}^{C \times P \times E}$. To capture the complex spatial dependencies inherent in this manifold, we leverage the Brain Topological Hierarchy (BTH) (Yang et al., 2025). The BTH organizes $F_{EEG}$ into a multi-level hierarchical structure, partitioning the features into $n$ topological levels denoted as $F_{BTH} = \{F^{B_1}, F^{B_2}, \dots, F^{B_n}\}$, ranging from the coarsest whole-brain level $F^{B_1}$ to the finest individual

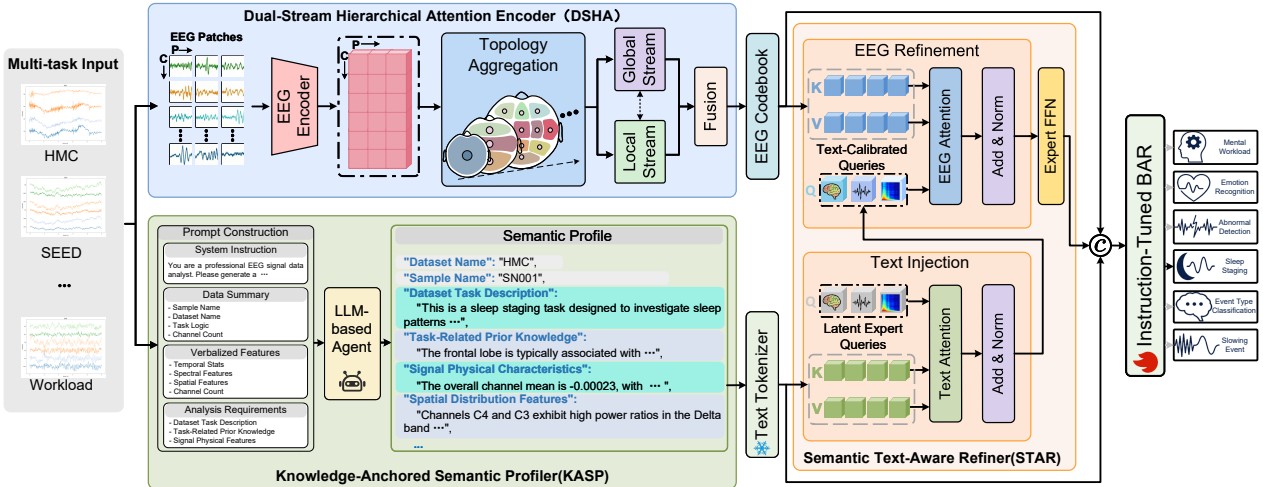

*Figure 2.* Overview of the **KAST-BAR** framework, featuring a **"Topological Encoding - Knowledge Anchoring - Guided Refinement"** pipeline. (1) **DSHA:** Encodes the non-Euclidean EEG manifold via bidirectional local-global interactions to produce topology-enhanced tokens. (2) **KASP:** Synthesizes signal-correlated medical profiles to anchor the model in expert clinical priors. (3) **STAR:** Leverages these profiles to guide Latent Expert Queries, enabling dynamic feature aggregation and robust cross-modal alignment.

electrode level $F^{B_n}$. Detailed topological hierarchy divisions are provided in Appendix C.1. Although prior works like THD-BAR utilize sequential hierarchical encoding, they typically propagate information unidirectionally. To address the issue of information loss in unidirectional propagation, we propose the Dual-Stream Hierarchical Attention (DSHA) encoder as Figure 3, which explicitly models bidirectional interactions through two interacted streams:

**Global Refinement Stream.** This stream is initialized with the coarsest representation $G_0 = F^{B_1}$ and iteratively integrates information from finer scales. At step $k$, it queries the next finer level $F^{B_{k+1}}$ via a Cross-Scale Attention(CSA) mechanism to refine global features using local details:

$$G'_k = \text{CSA}(Q = G_{k-1}, K = F^{B_{k+1}}, V = F^{B_{k+1}}) \quad (1)$$

$$G_k = \text{FFN}(\text{LN}(G'_k + G_{k-1})) \quad (2)$$

where $k = 1, 2, \ldots, n-1$, $\text{LN}(\cdot)$ denotes the Layer Normalization operator and $\text{FFN}(\cdot)$ represents the Feed-Forward Network.

**Local Context Stream.** Conversely, this stream starts with the finest representation $L_0 = F^{B_n}$ and queries coarser scales. At step $k$, it attends to the preceding coarser level $F^{B_{n-k}}$ to disambiguate local signals using global context:

$$L'_k = \text{CSA}(Q = L_{k-1}, K = F^{B_{n-k}}, V = F^{B_{n-k}}) \quad (3)$$

$$L_k = \text{FFN}(\text{LN}(L'_k + L_{k-1})) \quad (4)$$

Following the cross-scale interactions, we apply Multi-Head Self-Attention (MSA) modules at the end of each stream for intra-level integration. This process is formally expressed as $H_n = \text{FFN}(\text{LN}(\text{MSA}(H_{n-1}) + H_{n-1}))$, where

$H \in \{G, L\}$ represents the global and local representations, respectively.

Finally, we concatenate the refined global and local views and fuse them with intermediate topological features to generate the comprehensive EEG embedding $H_{EEG}$:

$$H_{EEG} = (G_n \oplus L_n) + \mathcal{F}\left(\{F^{B_i}\}_{i=2}^{n-1}\right) \quad (5)$$

where $\oplus$ denotes the concatenation operation along the feature dimension, and $\mathcal{F}(\cdot)$ represents a fusion module (e.g., a weighted sum or a convolution layer) that aggregates the set of intermediate features $\{F^{B_i}\}$ extracted from the 2-nd to the $(n-1)$-th level. To bridge the modality gap, we introduce a discretization module inspired by VQ-VAE (Van Den Oord et al., 2017) following the DSHA encoder. We maintain a learnable EEG Codebook $\mathcal{V}_{EEG} = \{v_i\}_{i=1}^{N_v}$. Each feature vector $h_i$ in $H_{EEG} \in \mathbb{R}^{N_e \times P \times E}$ is mapped to the index of its nearest neighbor in the codebook, yielding the discrete token sequence $S_{EEG}$:

$$S_{EEG} = \{k_i\}_{i=1}^{N_e \times P}, \quad k_i = \underset{j \in \{1,2,\ldots,N_v\}}{\arg\min} \|h_i - v_j\|_2 \quad (6)$$

Subsequently, the quantized latent vectors $Z_q$ are retrieved for subsequent processing:

$$Z_q = \{v_{k_i}\}_{i=1}^{N_e \times P} \quad (7)$$

This quantization strategy aligns continuous EEG signals with the textual vocabulary, enabling unified autoregressive processing. Detailed module implementation and pseudocode can be found in Appendix C.1.

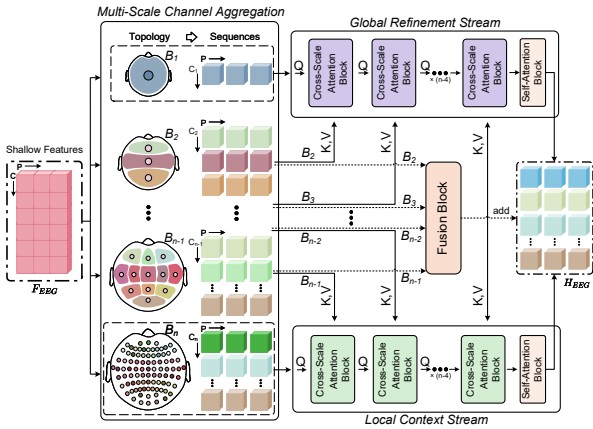

*Figure 3.* **Architecture of the DSHA Mechanism.** Unlike unidirectional approaches, DSHA introduces a interacted dual-stream structure based on the Brain Topology Hierarchy. It synergizes a **Global Refinement Stream** and a **Local Context Stream** through iterative Cross-Scale Attention. The resulting features are fused and quantized to align continuous EEG signals with the discrete textual vocabulary.

## 2.2. Knowledge-Anchored Semantic Profiler

To bridge the modality gap and endow the model with personalized semantic understanding, KASP adopts a label-free feature injection mechanism based on hard prompts. As illustrated in Figure 8, it utilizes a frozen LLM to transform EEG signals into expert-level semantic profiles rich in medical priors. This process consists of two stages:

**Physical Feature Extraction and Verbalization.** Given an EEG sample $X \in \mathbb{R}^{C \times T}$, we employ a set of deterministic operators $\Phi = \{\phi_{stat}, \phi_{spec}, \phi_{spat}\}$ to extract spectral, temporal, and spatial topological features, respectively (see Appendix C.2 for operator details). By jointly applying these operators, we map $X$ to a multi-dimensional physical feature vector $V_{phy}$:

$$V_{phy} = \Phi(X) = [\phi_{stat}(X), \phi_{spec}(X), \phi_{spat}(X)] \quad (8)$$

Subsequently, a verbalizer function $Verb(\cdot)$ is utilized to transform the numerical vector $V_{phy}$ into a structured sequence of natural language descriptions $S_{desc}$, achieving a preliminary alignment from signal space to linguistic space.

**Internal Knowledge Activation and Generation.** To activate the internal medical priors of the LLM, we construct a hard prompt $\mathcal{P}$ that strictly excludes classification labels. This prompt is dynamically concatenated from a task-specific description $\mathcal{I}_{task}$ (see Appendix D for detailed prompt templates) and the physical feature description $S_{desc}$:

$$\mathcal{P} = \mathcal{I}_{task} \oplus S_{desc} \quad (9)$$

where $\mathcal{I}_{task}$ contains data summaries and expert role settings. Finally, $\mathcal{P}$ is input into the frozen large language

model $\mathcal{M}_{LLM}$. By analyzing the association between features and the task, the model "awakens" its parametric medical knowledge to generate the final semantic profile $S_{text}$:

$$S_{text} = \mathcal{M}_{LLM}(\mathcal{I}_{task} \oplus Verb(\Phi(X))) \quad (10)$$

Surpassing simple data regurgitation, $S_{text}$ serves as an expert-level personalized analysis derived from the LLM's parametric knowledge. Crucially, this generative paradigm circumvents the reliance on scarce paired clinical reports required by ELM (Gijsen & Ritter, 2025), while offering fine-grained, instance-specific semantic profiles—a significant advancement over the static task-level instructions employed in UniMind (Lu et al., 2025). By encapsulating logical reasoning and medical context, $S_{text}$ provides high-level semantic anchors that guide the subsequent STAR module and enhance the BAR backbone's comprehension of underlying physiological dynamics.

## 2.3. Semantic Text-Aware Refiner

To effectively bridge the modality gap between KASP-generated semantic priors and DSHA-encoded physiological features, drawing inspiration from Q-Former (Li et al., 2023; Dai et al., 2023), we propose the Semantic Text-Aware Refiner (STAR). Moving beyond static concatenation, STAR introduces a query-based dynamic mechanism that utilizes a set of learnable latent experts to absorb task knowledge and actively retrieve and aggregate relevant physiological patterns. This process comprises three stages (see Appendix C.2 for detailed pseudo-code):

**Expert Calibration.** We initialize a set of learnable latent vectors $\mathcal{Q}_{lat} \in \mathbb{R}^{N_s \times E}$ as expert prototypes. A frozen text encoder is used to project the KASP-generated text $S_{text}$ into $H_{text} \in \mathbb{R}^{L \times E}$. Subsequently, semantic priors are injected into these latent experts via Multi-Head Cross-Attention (MHCA):

$$\mathcal{Q}_{calib} = \text{MHCA}(Q=\mathcal{Q}_{lat}, K=H_{text}, V=H_{text}) \quad (11)$$

where $\mathcal{Q}_{calib}$ denotes the text-calibrated expert queries, enabling them to adaptively attend to specific semantics within the textual descriptions.

**Semantically Guided Feature Aggregation.** The calibrated experts $\mathcal{Q}_{calib}$ serve as dynamic probes to extract task-relevant information from the DSHA-encoded discrete EEG representations $Z_q$. Through a second cross-attention step, the experts perform a weighted aggregation of signal features:

$$O_{STAR} = \text{MHCA}(Q=\mathcal{Q}_{calib}, K=Z_q, V=Z_q) \quad (12)$$

This operation condenses the high-dimensional data stream into compact and semantically rich summary tokens $O_{STAR} \in \mathbb{R}^{N_s \times E}$.

**Expert Projection and Orthogonality.** The aggregated features are processed via a feed-forward network to align with the LLM feature space, yielding the final output $S_{sem}$:

$$S_{sem} = \text{LN}(O_{STAR} + \text{FFN}(O_{STAR}))\tag{13}$$

To prevent mode collapse, we introduce an expert orthogonality loss $\mathcal{L}_{orth}$ to encourage diversity among experts:

$$\mathcal{L}_{orth} = \left\| \frac{\mathcal{Q}_{lat}\mathcal{Q}_{lat}^T}{\|\mathcal{Q}_{lat}\|_F^2} - I \right\|_F\tag{14}$$

## 3. Training Paradigm and Experimental Setup

In this section, we present the systematic training paradigm and the comprehensive experimental framework of KAST-BAR. We first detail the progressive learning pipeline, encompassing the self-supervised VQ reconstruction (Stage 1), the joint autoregressive pre-training (Stage 2), and the multi-task instruction fine-tuning. Subsequently, we introduce the experimental setup, including the construction of a large-scale pre-training corpus comprising 21 datasets, the selection of six diverse downstream benchmarks, and the specific implementation details regarding model configurations and training protocols.

### 3.1. Two-Stage Pre-training

**Stage 1: Self-Supervised VQ Reconstruction.** We jointly train the DSHA encoder and a learnable codebook to capture robust spatiotemporal topological representations. The continuous latent vectors $H_{EEG}$ are discretized into tokens $S_{EEG}$ and are used to reconstruct the original signals $\hat{X}_{EEG}$ and $\hat{X}_{fre}$ via two decoders. The optimization objective $\mathcal{L}_{DSHA}$ combines the time-frequency reconstruction loss with the quantization commitment loss:

$$\mathcal{L}_{DSHA} = \underbrace{\|X_{EEG} - \hat{X}_{EEG}\|_2^2 + \|X_{fre} - \hat{X}_{fre}\|_2^2}_{\mathcal{L}_{rec}}$$
$$+ \underbrace{\|\text{sg}[H_{EEG}] - Z_q\|_2^2 + \beta\|H_{EEG} - \text{sg}[Z_q]\|_2^2}_{\mathcal{L}_{quant}}\tag{15}$$

where $X_{fre}$ represents the frequency domain data obtained via DFT, and sg$[\cdot]$ denotes the stop-gradient operator. This stage yields a high-quality discrete tokenizer for EEG.

**Stage 2: Joint Autoregressive Pre-training.** We freeze the parameters of the DSHA encoder, expand the vocabulary of the Large Language Model (LLM) ($\mathcal{V}_{total} = \mathcal{V}_{text} \cup \mathcal{V}_{EEG}$), and integrate the KASP and STAR modules along with the LoRA-adapted BAR backbone. The model is trained on a hybrid sequence comprising domain knowledge text $S_{text}$, semantically-guided EEG aggregated features $S_{sem}$, and EEG tokens $S_{EEG}$. The primary Next Token Prediction

(NTP) loss encompasses both text and EEG modalities:

$$\mathcal{L}_{\text{NTP}} = \underbrace{-\frac{1}{|S_{text}|}\sum_{t=1}^{|S_{text}|}\log P(x_t^{text}|x_{<t}^{text})}_{\text{Text Loss}}$$
$$\underbrace{-\frac{1}{|S_{EEG}|}\sum_{k=1}^{|S_{EEG}|}\log P(x_k^{EEG}|S_{text}, S_{sem}, x_{<k}^{EEG})}_{\text{EEG Loss}}\tag{16}$$

where $x_t^{text} \in S_{text}$ represents the $t$-th token in the text sequence, and $x_k^{EEG} \in S_{EEG}$ represents the $k$-th token in the discrete EEG feature sequence. To prevent mode collapse among the latent experts in the STAR module, we introduce an orthogonality penalty. The final optimization objective is:

$$\mathcal{L}_{\text{CPT}} = \mathcal{L}_{\text{NTP}} + \lambda_{\text{orth}}\mathcal{L}_{\text{Orth}}\tag{17}$$

where $\lambda_{\text{orth}}$ denotes the weighting coefficient for the expert orthogonality loss. Minimizing $\mathcal{L}_{\text{CPT}}$ enables the model to effectively bridge the modality gap by leveraging medical prior knowledge and dynamic signal context.

### 3.2. Multi-task Instruction Fine-tuning

In the final stage, we adopt a multi-task instruction tuning paradigm to fine-tune the model across six downstream EEG tasks. We employ a decoupled fine-tuning strategy for the BAR backbone and the STAR module: a brand-new LoRA adapter is initialized after integrating the pre-trained BAR weights, while the STAR encoder undergoes full-parameter fine-tuning with a small learning rate multiplier. More detailed fine-tuning configurations are provided in Appendix C.4. Training is supervised via a masked instruction tuning loss, computed solely on the predicted label tokens of the response, with gradients for the instruction prompts and remaining input context $\mathcal{S}$ being masked out. The objective function is formulated as:

$$\mathcal{L}_{\text{SFT}} = -\frac{1}{|S_{ans}|}\sum_{t \in S_{ans}}\log P(a_t|\mathcal{S}, S_{ins}, a_{<t})\tag{18}$$

where $S_{ans}$ denotes the set of indices for the answer tokens, $S_{ins}$ represents the indices of the task instruction tokens, and $\mathcal{S}$ refers to the hybrid context sequence from the pre-training stage.

### 3.3. Datasets

**Pre-training Corpus.** To learn universal EEG representations capable of capturing complex physiological semantics, we constructed a large-scale pre-training corpus comprising 21 diverse EEG datasets. As detailed in Appendix Table 5, this extensive collection of more than 1,600 subjects covers

*Table 1.* Summary of the 6 EEG datasets used for downstream instruction fine-tuning.

| Task | Dataset | Subject | Channel | Rate | Label |
|------|---------|---------|---------|------|-------|
| Workload | Workload | 36 | 19 | 500Hz | 2 |
| Emotion | SEED | 15 | 62 | 1000Hz | 3 |
| Abnormal | TUAB | 2383 | 23 | 256Hz | 2 |
| Sleep | HMC | 151 | 4 | 256Hz | 5 |
| Event | TUEV | 370 | 23 | 256Hz | 6 |
| Slowing | TUSL | 28 | 23 | 256Hz | 3 |

a wide range of paradigms, varying sampling rates, and channel configurations, providing a robust foundation for the self-supervised learning of the DSHA encoder and the KASP-guided STAR module.

**Downstream Tasks.** To rigorously evaluate the generalization capabilities of KAST-BAR in instruction-following scenarios, we selected 6 distinct datasets covering representative EEG applications. These tasks include Mental Workload Recognition (Workload (Zyma et al., 2019)), Emotion Recognition (SEED (Zheng & Lu, 2015)), Abnormal Detection (TUAB (Harati et al., 2015)), Sleep Staging (HMC (Alvarez-Estevez & Rijsman, 2022)), Event Type Classification (TUEV (Harati et al., 2015)), and Slowing Event Classification (TUSL (von Weltin et al., 2017)). A summary of these datasets is presented in Table 1.

### 3.4. Implementation Details

**Data Preprocessing.** Following standard pipelines(Jiang et al., 2025; Yang et al., 2025), we unified signal characteristics by re-sampling all EEG signals to 200 Hz. We applied a 0.1–75 Hz band-pass filter and a 50/60 Hz notch filter to remove noise and power-line interference. Finally, we utilized Interquartile Range (IQR) based Robust Scaling to ensure stable normalization against artifacts.

**Model Configurations.** We developed two variants, **KAST-BAR-Base** (0.8B) and **KAST-BAR-Large** (2.2B). In the DSHA module, the EEG encoder and decoder adopt the vanilla Transformer (Vaswani et al., 2017) architecture consistent with THD-BAR (Yang et al., 2025), utilizing a five-scale Brain Topology Hierarchy ($B_1$−$B_5$; details in Appendix C.1) and a patch size of $W = 200$. The KASP module employs a frozen Qwen2.5-7B. The STAR refiner is configured with 8 attention heads and $N_s = 16$ (Base) or $N_s = 32$ (Large) learnable latent queries, with embedding dimensions $E$ aligned to the respective BAR backbones.

**BAR Backbone.** Unlike previous works utilizing the GPT-2 series (Jiang et al., 2025; Yang et al., 2025), we employ the more powerful **Qwen2.5-0.5B/1.5B** series (Yang et al., 2024) as our backbone. The model is trained and fine-tuned using Low-Rank Adaptation (LoRA) (Hu et al., 2022) applied to all linear projection layers. Detailed hyperparameter settings are provided in Appendix C.3.

**Training Protocols.** Our experiments were conducted on a high-performance computing cluster equipped with 8 NVIDIA L40s-48G GPUs, using Python 3.12.11. Further details regarding training and fine-tuning settings are comprehensively described in Appendix C.3 C.4.

## 4. Experimental Results

### 4.1. Baselines and Evaluation Metrics

**Baselines.** To comprehensively assess the efficacy of KAST-BAR, we benchmark it against a range of leading methods. These baselines are categorized into *Single-Task Models* and *Multi-Task Models* based on their training (or fine-tuning) paradigms on downstream datasets. **Single-Task Baselines:** We select classic non-generic models such as EEG-Net (Lawhern et al., 2018) and SPaRCNet (Jing et al., 2023). Additionally, we compare against recent general-purpose foundation models that are fine-tuned in a single-task manner, including LaBraM (Jiang et al., 2024), CBraMod (Wang et al., 2025a), and CSBrain (Zhou et al., 2025). **Multi-Task Baselines:** We benchmark against leading multi-task frameworks, including EEGPT (Wang et al., 2024), NeuroLM (Jiang et al., 2025), and THD-BAR (Yang et al., 2025).

**Evaluation Metrics.** To rigorously evaluate performance on class-imbalanced physiological datasets, we adopt **Balanced Accuracy (B-Acc)** as the primary metric across all tasks. Furthermore, we complement this with task-specific metrics: for *binary classification*, we report **AUROC** and **AUC-PR** to assess threshold-independent discriminative capability; for *multi-class classification*, we utilize **Cohen's Kappa** ($\kappa$) and **F1-Score** to measure prediction agreement and the balance between precision and recall. Detailed mathematical formulations and implementation details are provided in Appendix E.

### 4.2. Downstream Tasks Performance

We conducted extensive experiments on the six diverse downstream datasets introduced previously. The quantitative results are summarized in Table 2. Additionally, to provide a more granular analysis, we present comprehensive metrics (e.g., AUC-PR, Kappa) for the TUAB and TUEV datasets in Table 3 as representative examples. Full detailed results for all remaining datasets are provided in Appendix F.

**Comparison with Baselines.** As shown in Table 2, our proposed KAST-BAR demonstrates superior generalization capabilities.

**(1) Against Multi-Task Models:** KAST-BAR-Large establishes a new benchmark among foundation models. It achieves substantial improvements over NeuroLM-XL

*Table 2.* **Balanced Accuracy (B-Acc)** comparison across six downstream datasets, where the best multi-task and single-task results are marked in **bold** and underlined, respectively.

| | Methods | Parameters | Workload | TUSL | TUAB | TUEV | SEED | HMC |
|---|---|---|---|---|---|---|---|---|
| *Single-Task* | EEGNet (Lawhern et al., 2018) | - | 60.9±2.6 | 57.4±4.2 | 77.1±0.6 | 39.8±1.1 | 62.5±3.5 | 62.2±2.1 |
| | SPaRCNet (Jing et al., 2023) | 0.79M | 59.8±0.7 | 56.9±2.5 | 77.5±1.0 | 43.2±3.2 | 56.0±2.4 | 55.4±3.7 |
| | ST-Transformer (Song et al., 2021) | 3.5M | 61.0±0.6 | 49.3±5.6 | 79.3±0.2 | 39.6±1.8 | 56.4±1.8 | 59.5±5.4 |
| | BIOT (Yang et al., 2023) | 3.2M | 66.6±1.1 | 57.6±3.0 | 79.6±0.6 | 52.8±2.2 | 71.0±0.2 | 68.6±0.4 |
| | LaBraM-Base (Jiang et al., 2024) | 5.8M | 66.1±2.0 | 76.3±2.3 | 81.4±0.2 | 64.1±0.7 | 73.2±0.2 | 72.9±1.0 |
| | CBraMod (Wang et al., 2025a) | 4.0M | 65.4±2.6 | 73.9±3.2 | 78.9±0.3 | 66.7±1.1 | 72.7±0.7 | 72.7±0.4 |
| | CSBrain (Zhou et al., 2025) | 4.9M | - | 85.7±2.4 | 81.7±0.4 | 69.0±0.6 | 73.0±0.4 | 73.5±0.5 |
| *Multi-Task* | EEGPT (Wang et al., 2024) | 25M | 63.0±1.8 | 72.9±1.4 | 79.2±0.4 | 62.3±1.1 | 71.2±0.2 | 70.3±0.8 |
| | NeuroLM-XL (Jiang et al., 2025) | 1.7B | 63.5±4.4 | 68.5±3.0 | 79.7±0.9 | 46.8±3.6 | 60.3±0.1 | 57.6±10.8 |
| | THD-BAR-Huge (Yang et al., 2025) | 1.6B | 67.1±5.8 | 69.2±2.3 | 82.2±0.4 | 65.3±0.5 | 73.9±0.3 | 68.4±4.3 |
| | **KAST-BAR-Base (Ours)** | 0.8B | 68.6±2.1 | 73.7±3.3 | 81.3±0.6 | 66.1±0.9 | 72.7±0.4 | 72.5±2.1 |
| | **KAST-BAR-Large (Ours)** | 2.2B | **71.2±1.7** | **77.4±3.9** | **83.2±1.1** | **70.8±1.4** | **74.3±0.2** | **73.9±1.7** |

*Table 3.* Performance comparison on TUAB and TUEV datasets, where the best multi-task and single-task results are marked in **bold** and underlined, respectively.

| | Methods | Model Parameter | TUAB | | | TUEV | | |
|---|---|---|---|---|---|---|---|---|
| | | | B-Acc | AUC-PR | AUROC | B-Acc | Kappa | F1-W |
| *Single-Task* | EEGNet (Lawhern et al., 2018) | - | 77.1±0.6 | 82.3±0.3 | 85.0±0.3 | 39.8±1.1 | 34.9±1.1 | 63.8±2.1 |
| | SPaRCNet (Jing et al., 2023) | 0.79M | 77.5±1.0 | 83.1±0.7 | 86.3±0.6 | 43.2±3.2 | 43.8±2.2 | 68.1±1.7 |
| | ST-Transformer (Song et al., 2021) | 3.5M | 79.3±0.2 | 85.4±0.6 | 86.9±0.2 | 39.6±1.8 | 38.3±2.3 | 69.1±2.1 |
| | BIOT (Yang et al., 2023) | 3.2M | 79.6±0.6 | 87.9±0.2 | 88.2±0.4 | 52.8±2.2 | 52.7±2.5 | 74.9±0.8 |
| | LaBraM-Base (Jiang et al., 2024) | 5.8M | 81.4±0.2 | 89.7±0.1 | 90.2±0.1 | 64.1±0.7 | 66.4±1.0 | 83.1±0.5 |
| | CBraMod (Wang et al., 2025a) | 4.0M | 78.9±0.3 | 86.4±0.6 | 86.1±0.6 | 66.7±1.1 | 67.7±1.0 | 83.4±0.6 |
| | CSBrain (Zhou et al., 2025) | 4.9M | 81.7±0.4 | 90.1±0.7 | 89.6±0.5 | 69.0±0.6 | 68.3±0.5 | 83.3±0.6 |
| *Multi-Task* | EEGPT (Wang et al., 2024) | 25M | 79.2±0.4 | 74.6±0.5 | 86.6±0.7 | 62.3±1.1 | 63.5±1.3 | 81.9±0.6 |
| | NeuroLM-XL (Jiang et al., 2025) | 1.7B | 79.7±0.9 | 72.2±0.8 | 78.8±1.9 | 46.8±3.6 | 45.7±5.0 | 73.6±2.2 |
| | THD-BAR-Huge (Yang et al., 2025) | 1.6B | 82.2±0.4 | 84.8±0.7 | 88.6±1.2 | 65.3±0.5 | 64.4±2.3 | 82.1±1.4 |
| | KAST-BAR-Base(Ours) | 0.8B | 81.3±0.6 | 85.6±0.5 | 88.2±1.4 | 66.1±0.9 | 64.7±1.8 | 82.8±1.5 |
| | KAST-BAR-Large(Ours) | 2.2B | **83.2±1.1** | **87.1±1.3** | **90.7±2.1** | **70.8±1.4** | **69.6±2.2** | **85.5±1.1** |

(1.7B), delivering gains of 7.7% on Workload and 14.0% on SEED. The comparison with the previously leading method, THD-BAR-Huge, is even more compelling. Despite sharing a comparable parameter scale of approximately 2B, KAST-BAR-Large consistently outperforms THD-BAR-Huge across all datasets, with notable improvements of 8.2% on TUSL and 5.5% on TUEV. Remarkably, even our significantly smaller KAST-BAR-Base model, with merely half the parameters, outperforms NeuroLM-XL on all tasks and surpasses THD-BAR-Huge on four datasets. This comprehensive superiority strongly validates that our KASP-guided semantic-topological orchestration mechanism captures complex bio-signals more effectively than the pure topological modeling relied upon by THD-BAR, proving that the performance gains stem from architectural innovation rather than mere model scaling.

**(2) Against Single-Task Models:** Remarkably, KAST-BAR achieves competitive or superior performance even when compared to models heavily optimized for single tasks. It achieves superior performance on 5 out of the 6 datasets. However, on the TUSL task, although KAST-BAR-Large

achieves a substantial improvement (77.4%) over the current best multi-task foundation model THD-BAR (69.2%), it still trails behind the specialized CSBrain model (85.7%). We attribute this to the fact that slow-wave events rely heavily on specific, fine-grained frequency-domain features. KAST-BAR, which utilizes multi-task fine-tuning to prioritize cross-task semantic alignment and topological spatial modeling, may be less sensitive to such specific spectral details compared to single-task models. For more detailed experimental results and analysis, see Appendix F.

**Model Scaling.** Consistent with findings in Large Language Models (LLMs)(Kaplan et al., 2020), we observe a clear Scaling Law during the pre-training stage. As illustrated in Figure 4, KAST-BAR-Large (2.2B) exhibits significantly faster convergence and a steeper loss reduction trajectory compared to the Base model (0.8B). This superior representation capability directly translates to downstream performance, where the Large model consistently outperforms the Base version across all fine-tuning tasks. This suggests that, when coupled with our effective semantic alignment strategy, scaling up the LLM backbone leads to both efficient

*Table 4.* Ablation study on the TUEV, SEED, and HMC dataset. The proposed KAST-BAR corresponds to row (e).

| ID | Tokenizer | STAR | KASP | TUEV | SEED | HMC |
|----|-----------|------|------|------|------|-----|
| (a) | VQ | ✓ | ✓ | 60.9 | 68.7 | 65.2 |
| (b) | THVQ | ✓ | ✓ | 67.4 | 74.0 | 70.6 |
| (c) | DSHA | ✗ | ✓ | 66.7 | 72.9 | 70.1 |
| (d) | DSHA | ✓ | ✗ | 68.0 | 73.4 | 72.5 |
| (e) | DSHA | ✓ | ✓ | **70.8** | **74.3** | **73.9** |

learning and robust physiological understanding.

### 4.3. Ablation Study

To validate the effectiveness and generalizability of our proposed components, we conducted ablation studies using the KAST-BAR-Large model across three diverse datasets: TUEV (event), SEED (emotion), and HMC (sleep). The results, summarized in Table 4 and Figure 4, verify the necessity and cross-task consistency of each module:

**DSHA vs. THVQ vs. VQ:** We replaced the DSHA encoder with the THVQ-VAE utilized in THD-BAR and a vanilla VQ-VAE, respectively. The training curves in Figure 4(a), combined with the quantitative results in Table 4, substantiate DSHA's capability to capture robust spatiotemporal features, achieving consistent performance gains ranging from +5.6% to +9.9% (vs. VQ) and +0.3% to +3.4% (vs. THVQ) across the three datasets. We attribute this improvement to its dynamic spatial attention mechanism; in contrast, THVQ-VAE imposes rigid topological constraints, while the vanilla VQ-VAE lacks any topological inductive bias.

**STAR vs. Static Backbone:** Removing the STAR refiner in favor of a static projection layer resulted in noticeably higher pre-training loss and subsequent performance degradation across all tasks, proving STAR's contribution with accuracy gains ranging from +1.4% to +4.1%. This demonstrates that the dynamic refinement mechanism is pivotal for multi-task learning, enabling the model to adaptively select relevant physiological features conditioned on textual priors.

**KASP vs. Fixed Descriptions:** Substituting the KASP-generated semantic profiles with simple dataset-level textual descriptions led to inferior loss convergence and a decrease in final accuracy, yielding performance declines of 0.9% to 2.8% across the three datasets. This verifies that the rich, sample-level expert medical knowledge injected by KASP provides a stronger semantic prior, bridging the modality gap more effectively than sparse, fixed textual descriptions.

### 4.4. Visualization & Interpretability

To elucidate the interpretability of our paradigm, we visualize the spatial attention of STAR experts in the KAST-BAR-Base model (Figure 5 and Figure 6). These heatmaps demonstrate how the KASP module orchestrates latent ex-

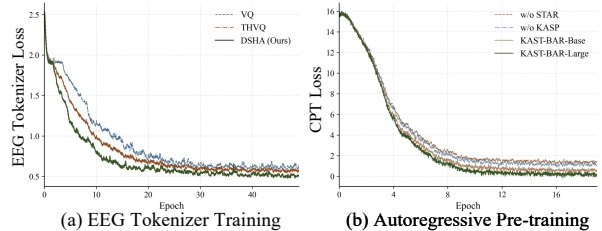

(a) EEG Tokenizer Training     (b) Autoregressive Pre-training

*Figure 4.* Training dynamics of KAST-BAR. (a) Reconstruction loss comparison between DSHA and baselines (VQ, THVQ); (b) CPT loss during autoregressive pre-training, comparing full models (Base/Large) with ablation variants (w/o STAR, w/o KASP).

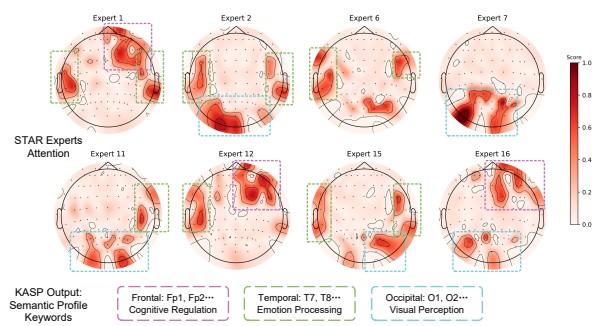

*Figure 5.* Visualization on **SEED (Emotion)**. Driven by **Visual-Emotional** semantic priors (bottom), latent experts (top) specifically cluster around **Frontal, Temporal and Occipital** regions, demonstrating knowledge-guided topological alignment.

perts to actively aggregate features based on comprehensive medical insights.

In the SEED emotion recognition task (Figure 5), guided by priors such as "Visual Perception" and "Cognitive Regulation," the experts establish a distributed functional connectivity network. Specifically, distinct experts (e.g., Expert 16) bridge the Occipital and Frontal lobes, effectively linking visual stimuli processing with high-level cognitive control. Simultaneously, others (e.g., Experts 1 and 12) span the Frontal and Temporal regions to facilitate emotional integration. This granularity reveals that STAR spontaneously decouples complex brain activity into task-specific topological subspaces. Conversely, the HMC task (Figure 6) exhibits a marked topological reconfiguration. Here, attention adaptively realigns to the Frontal-Central zones—the physiological generation sites of Sleep Spindles—while specific experts isolate Occipital Alpha rhythms to detect wakefulness patterns. This dynamic aggregation of brain topology validates our paradigm, demonstrating that the model transcends static geometric constraints to learn robust, interpretable cross-modal representations. Comprehensive visualization analysis of the remaining datasets is detailed in Appendix G.

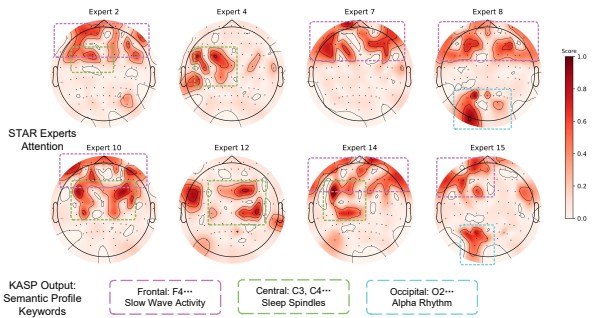

*Figure 6.* Visualization on **HMC (Sleep)**. In contrast to SEED, latent experts (top) here adaptively shift focus to **Frontal, Central and Occipital** regions, anchored by **Sleep-specific** semantic priors (bottom) such as spindles and delta waves.

## 5. Conclusion

In this paper, we introduced KAST-BAR, a foundation model synergizing dynamic topological perception with knowledge-driven reasoning. By integrating the DSHA encoder and the KASP-STAR mechanism, we established an explicit "Topological Encoding - Knowledge Anchoring - Guided Refinement" paradigm. This approach bridges the modality gap to yield robust, interpretable cross-modal representations, overcoming the limitations of existing alignment strategies. Extensive validation across 21 datasets and 6 downstream tasks confirms KAST-BAR's superior generalization, laying a solid foundation for clinical diagnosis and next-generation universal Brain-Computer Interfaces.

## Acknowledgements

This work was supported in part by the National Natural Science Foundation of China under Grant U24B20186, Grant 62325301, Grant 32541016; in part by the Beijing Natural Science Foundation under Grant Z220017, Grant L256008; in part by the National Key Research and Development Program of China under Grant 2023YFC2416600; in part by the National Key Research and Development Program of China under Grant 2024YFC3606900, Grant 2024YFC3606903. (Corresponding author: Yang Li)

## Impact Statement

This paper introduces KAST-BAR, a foundation model framework designed to align EEG signals with medical knowledge and large language models. The primary goal of this work is to advance the development of interpretable and robust BCI systems for clinical applications, such as automated sleep staging and neurological anomaly detection. By reducing the reliance on massive labeled data for individual subjects, our method has the potential to democratize access to high-quality neurological analysis.

However, we acknowledge the potential societal implica-

tions associated with integrating LLMs into healthcare. While our Knowledge-Anchored Semantic Profiler (KASP) and guided refinement mechanisms are designed to ground the model's outputs in expert medical knowledge, the risk of generation hallucinations inherent to LLMs cannot be entirely eliminated. Therefore, KAST-BAR is intended as an assistive tool for clinicians rather than an autonomous diagnostic system. Furthermore, as our model performs effectively on emotion recognition and workload assessment tasks, we strongly advocate for strict ethical guidelines to prevent the misuse of such technology for non-consensual neuro-surveillance or privacy infringement. We encourage researchers and practitioners to prioritize data privacy and user consent when deploying these models in real-world scenarios.

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

# A. Related Work

**From Task-Specific Architectures to Universal Foundation Models.** The fundamental impetus for developing EEG foundation models lies in constructing "universal neural representations" capable of spanning diverse neurological tasks. Traditional deep learning methodologies often succumb to an isolated "one-task-one-architecture" paradigm. For instance, EEGNet (Lawhern et al., 2018) employs compact depthwise separable convolutions to optimize BCI decoding efficiency; SPaRCNet (Jing et al., 2023) utilizes densely connected convolutional networks for precise seizure and abnormal waveform identification; and ST-Transformer (Song et al., 2021) introduces spatiotemporal attention to enhance emotion recognition accuracy. Although these specialized models excel in their respective isolated domains, they rely heavily on supervision signals from single datasets, making effective transfer across neurological tasks difficult. Furthermore, they often overlook the deep spatiotemporal dynamic mechanisms shared by brain activities across different pathological states. This fragmented modeling approach severely hinders the scalable deployment of Brain-Computer Interface (BCI) technologies in complex clinical environments. In contrast, EEG foundation models aim to learn the intrinsic syntax and spatiotemporal topology of brain activities through pre-training on large-scale, diverse, unlabeled corpora. This universal representation not only adapts to a wide range of downstream tasks via lightweight fine-tuning but, more importantly, promises to capture biologically interpretable deep neural patterns beyond the limitations of single datasets, providing a unified computational substrate for automated neurological diagnosis and human-machine interaction.

**Self-Supervised Learning for EEG Representation.** The transformative success of the Self-Supervised Learning (SSL) paradigm in NLP and CV (Devlin et al., 2019; He et al., 2022) has catalyzed its application in the neurophysiological domain to overcome the scarcity of labeled data. Pioneering works such as LaBraM (Jiang et al., 2024) and NeuroLM (Jiang et al., 2025) leverage Transformer architectures, treating EEG signals as flattened 1D sequences or 2D time-frequency images, and utilize Masked Autoencoder (MAE) or Autoregressive (AR) objectives to capture long-range dependencies. Although these methods have verified the Scaling Laws in neural data, they inherently neglect the non-Euclidean properties of brain signals, leading to a severe **"Topological Mismatch."** To address this, recent works have begun to explicitly introduce spatial priors: Uni-NTFM (Chen et al., 2026) adopts a multi-scale tokenization strategy to aggregate regional information, while THD-BAR (Yang et al., 2025) simulates the brain's physical connectivity through hierarchical topological constraints. These advancements mark an evolution in EEG representation learning from pure temporal modeling to spatiotemporal topological modeling that better aligns with neurophysiological essence.

**Cross-Modal Alignment and Semantic Reasoning.** The cross-modal alignment paradigm first achieved breakthroughs in the vision-language domain (e.g., CLIP (Radford et al., 2021) and BLIP-2 (Li et al., 2023)), enabling powerful zero-shot transfer capabilities by constructing a shared semantic manifold. Inspired by this, the neurophysiological computing field has seen the emergence of four main alignment technical routes aimed at bridging the gap between EEG signals and human language:

- **Alignment with General Open-Domain Corpora**: Early attempts like NeuroLM (Jiang et al., 2025) directly utilized the general semantic space of pre-trained LLMs (e.g., GPT) to constrain EEG representations. However, unlike natural images, there exists an intrinsic **"Semantic Dissonance"** between EEG signals and general text. Due to the lack of domain knowledge anchoring, such forced spatial alignment often leads to representation space collapse, failing to capture physiologically meaningful patterns.

- **Contrastive Learning with Static Descriptions**: Drawing on CLIP, DistillCLIP (Wang et al., 2025b) and LEAF (Jiang et al., 2026) focus on aligning EEG embeddings with predefined text labels. LEAF further introduces a task-level instruction query mechanism to bridge the semantic distance. While empowering models with zero-shot capabilities, these methods remain limited by the **static and coarse-grained nature** of predefined text prompts, making it difficult to precisely capture the fine-grained, continuously changing physiological dynamics inherent in signals.

- **Alignment via Clinical Reports**: ELM (Gijsen & Ritter, 2025) leverages clinical expert diagnostic reports as rich supervisory signals, enabling the model to comprehend the prior knowledge embedded within clinical narratives. Despite offering comprehensive semantic details, its scalability is severely constrained by the **extreme scarcity** of high-quality paired EEG-report data and medical privacy regulations, making it difficult to replicate the data scaling effects observed in VLMs.

- **Instruction Prompting with Data Embeddings**: Recent research such as SP-LLM (Zhao et al., 2025) explores converting EEG into textualized data representations (e.g., statistical features) and inputting them directly into LLMs as prompts. This approach attempts to leverage LLM reasoning capabilities but faces the limitation of **"shallow**

**numerical mapping.''** Lacking domain-relevant deep priors, LLMs often struggle to establish deep causal connections between text descriptions and raw EEG data, failing to truly realize deep reasoning from physical features to medical semantics.

Currently, the core challenge in this field remains how to fundamentally resolve the "Semantic Dissonance" between general language and physiological features, and how to effectively handle significant distributional heterogeneity across subjects and tasks.

**Dynamic Neural Networks and Mixture-of-Experts (MoE).** To address the distributional heterogeneity of EEG data, dynamic neural networks have gradually become a research hotspot. The Mixture-of-Experts (MoE) architecture (Shazeer et al., 2017) offers an efficient solution for multi-task learning through the conditional activation of network sub-modules via gating mechanisms. This trend is particularly evident in the cross-modal domain: for instance, in image processing, Q-former (Li et al., 2023) introduced a query-based dynamic mechanism capable of adaptively extracting relevant information from visual features based on text conditions, demonstrating the superiority of dynamic aggregation.

Turning to the EEG domain, although LEAF (Jiang et al., 2026) initially introduced a semantic instruction query mechanism to achieve a leap from "static alignment" to **"dynamic generation,"** introducing dynamism only at the alignment stage is insufficient when facing highly complex heterogeneous data. Recently, UniMind (Lu et al., 2025) has begun exploring MoE structures to handle diverse downstream tasks, attempting to construct a dynamic backbone adaptable to different task features. However, traditional MoE often relies on **"Sparse Routing"** (i.e., hard selection of Top-$k$ experts). This mechanical dynamic selection is prone to causing the fragmentation of critical spatiotemporal information when processing continuous and holistic physiological signals. Therefore, the central challenge of current research lies in utilizing contextual priors to **adaptively modulate** dynamic aggregation weights, thereby achieving precise, dynamic capture of heterogeneous physiological features without losing **"Holistic Contextual Information."**

## B. Datasets and Data Preparation Details

In this section, we provide detailed specifications of the datasets utilized in our experiments. To construct a robust foundation model, we aggregated a large-scale pre-training corpus comprising 21 public EEG datasets, covering diverse paradigms such as medical diagnosis, sleep staging, emotion recognition, and brain-computer interfaces (BCI). For downstream evaluation, we selected 6 representative datasets to assess the model's adaptability to specific instructions.

### B.1. Pre-training Corpus

The pre-training corpus comprises EEG recordings from approximately 1,600 subjects. We standardized the data by re-sampling all signals to 200 Hz and unifying channel configurations. Table 5 summarizes the key parameters of the 21 datasets included in the pre-training phase.

### B.2. Downstream Tasks

We fine-tuned KAST-BAR on 6 downstream datasets representing distinct EEG analysis domains. Table 6 details the specific configurations for each fine-tuning task.

## C. Model Architecture and Implementation Details

In this section, we provide the granular implementation details, algorithmic descriptions, and hyperparameter configurations necessary for reproducing KAST-BAR.

### C.1. Detailed Architecture of DSHA Encoder & Quantizer

**1. Brain Topology Hierarchy (BTH) Specification**: Adhering to the 5-scale BTH architecture in THD-BAR (Yang et al., 2025), we organize electrodes into a coarse-to-fine hierarchy ($B_1 \rightarrow B_5$) based on spatial proximity and anatomical divisions, as illustrated in Figure 7. Specifically, **Level 1 (Whole Brain, $B_1$)** treats all channels as a single global unit to capture widespread synchronous events; **Level 2 (Major Regions, $B_2$)** parcellates the brain into three primary *horizontal bands*—Anterior (Frontal), Central (Sensorimotor/Parietal), and Posterior (Occipital/Temporal)—to reflect macro-scale functional specializations; **Level 3 (Sub-regions, $B_3$)** further subdivides these horizontal bands into lateralized zones (e.g.,

*Table 5.* Summary of the 21 EEG datasets used for self-supervised pre-training. The collection covers a wide range of tasks, sampling rates, and channel configurations to ensure the diversity of the learned representations.

| Dataset | Subject | Channel | Rate (Hz) | Description |
|---|---|---|---|---|
| *Clinical & Epilepsy* | | | | |
| TUH Seizure (TUSZ) (Shah et al., 2018) | 675 | 19-23 | 256 | Seizure detection with precise event start/stop annotations. |
| TUH Artifact (TUAR) (Buckwalter et al., 2021) | 200+ | 23 | 256 | Clinical EEG dataset annotated for 5 types of artifacts. |
| TUH Epilepsy (TUEP) (Veloso et al., 2017) | 200 | 19-23 | 256 | Epilepsy diagnosis dataset verified by expert neurologists. |
| CHB-MIT (Goldberger et al., 2000; Shoeb, 2009) | 21 | 16 | 256 | Pediatric scalp EEG recordings with seizure annotations. |
| Siena Scalp EEG (Detti et al., 2020) | 14 | 31 | 512 | Long-term monitoring data covering diverse clinical states. |
| *Emotion Recognition* | | | | |
| SEED-IV (Zheng et al., 2018) | 15 | 62 | 1000 | 4-class emotion recognition (happy, sad, fear, neutral). |
| SEED-V (Liu et al., 2021) | 15 | 62 | 1000 | 5-class emotion recognition task with video stimuli. |
| SEED-GER (Liu et al., 2022) | 8 | 62 | 1000 | Emotion recognition dataset collected from German subjects. |
| SEED-FRA (Liu et al., 2022) | 8 | 62 | 1000 | Emotion recognition dataset collected from French subjects. |
| DEAP (Koelstra et al., 2011) | 32 | 32 | 128 | Multimodal dataset for valence-arousal affect analysis. |
| EmoBrain (Savran et al., 2006) | 16 | 64 | 1024 | Multimodal dataset for emotion detection. |
| *BCI, Cognitive & Other* | | | | |
| Grasp and Lift (Luciw et al., 2014) | 12 | 32 | 500 | Detection of 6 hand movement events (Grasp-and-Lift task). |
| Inria BCI Challenge (Margaux et al., 2012) | 12 | 32 | 500 | BCI challenge dataset for error potential or event detection. |
| BCIC IV 1 (Blankertz et al., 2007) | 7 | 59 | 1000 | Motor imagery tasks with continuous EEG signals. |
| BCIC IV 2 (Brunner et al., 2008) | 9 | 22 | 250 | 4-class motor imagery (left/right hand, feet, tongue). |
| EEGMMI (Schalk et al., 2004) | 109 | 64 | 160 | Large-scale motor movement and imagery database (PhysioNet). |
| Raw EEG Data (Trujillo, 2020) | 30 | 64 | 256 | Raw EEG recordings for general sensory and cognitive analysis. |
| Resting State EEG Data (Trujillo et al., 2017) | 22 | 64 | 256 | Resting state recordings from healthy subjects (eyes open/closed). |
| SPIS Resting State (Torkamani-Azar et al., 2020) | 10 | 64 | 2048 | High-sampling rate resting state data (eyes open/closed). |
| Target Versus Non-Target (Korczowski et al., 2019) | 50 | 32 | 512 | Visual P300 "Brain Invaders" paradigm for target detection. |
| STEW (Lim et al., 2018) | 45 | 14 | 128 | Simultaneous Task EEG Workload assessment dataset. |

*Table 6.* Detailed specifications of the 6 downstream fine-tuning datasets.

| Task | Dataset | Label | Channel | Rate (Hz) | Task Description |
|------|---------|-------|---------|-----------|------------------|
| Workload | Workload (Zyma et al., 2019) | 2 | 19 | 500 | Binary classification of mental arithmetic load vs. rest. |
| Emotion | SEED (Zheng & Lu, 2015) | 3 | 62 | 1000 | Recognition of positive, neutral, and negative emotional states. |
| Abnormal | TUAB (Harati et al., 2015) | 2 | 23 | 256 | Clinical anomaly detection (Normal vs. Abnormal). |
| Sleep | HMC (Alvarez-Estevez & Rijsman, 2022) | 5 | 4 | 256 | 5-stage sleep scoring (W, N1, N2, N3, REM). |
| Event | TUEV (Harati et al., 2015) | 6 | 23 | 256 | Classification of 6 event types (e.g., SPSW, GPED, PLED). |
| Slowing | TUSL (von Weltin et al., 2017) | 3 | 23 | 256 | Detection of electroencephalographic slowing events. |

splitting the Anterior band into Left-, Mid-, and Right-Frontal areas) to capture inter-regional interplay; this is followed by **Level 4 (Channel Clusters, $B_4$)** representing fine-grained local neighborhoods, and finally **Level 5 (Individual Channels, $B_5$)** corresponding to original discrete electrodes. This hierarchical definition serves as the structural basis for initializing the Global Refinement Stream ($G_0 \leftarrow B_1$) and the Local Context Stream ($L_0 \leftarrow B_5$) in the DSHA encoder.

**2. Vector Quantizer Configuration.** We employ the standard VQ-VAE (Van Den Oord et al., 2017) mechanism to discretize the multi-channel continuous EEG signals output by the DSHA encoder. Detailed network architecture specifications are provided in Table 7.

**3. DSHA Structure: Dual-Stream Interaction Mechanism.** Going beyond simple unidirectional multi-scale feature aggregation, DSHA adopts an explicit **Bidirectional Progressive Interaction Strategy** to address the dilution of local details or the absence of global context. Specifically, the model maintains two parallel processing streams:

- **Global Refinement Stream:** Initialized from the coarsest whole-brain representation ($G_0 = F^{B_1}$). It simulates a **"Zoom-in"** process, where at each iteration step, the macroscopic view actively queries features from the next finer level ($F^{B_{k+1}}$) via Cross-Scale Attention. This process allows the model to enrich and refine global features using locally specific signal patterns.

- **Local Context Stream:** Initialized from the finest individual electrode representation ($L_0 = F^{B_5}$). It simulates a **"Zoom-out"** process, where the microscopic view progressively integrates information from coarser scales ($F^{B_{k-1}}$). This mechanism serves a dual purpose: first, it leverages broad regional consistency to suppress local atypical artifacts; second, it anchors isolated local features within the background of global neural activity, revealing the intrinsic dependencies between micro-scale activities and the macro-scale brain network.

Upon completing cross-scale interactions, we apply intra-layer self-attention to each stream for feature smoothing. Finally, the outputs of both streams are fused with intermediate hierarchical features to generate a comprehensive EEG representation

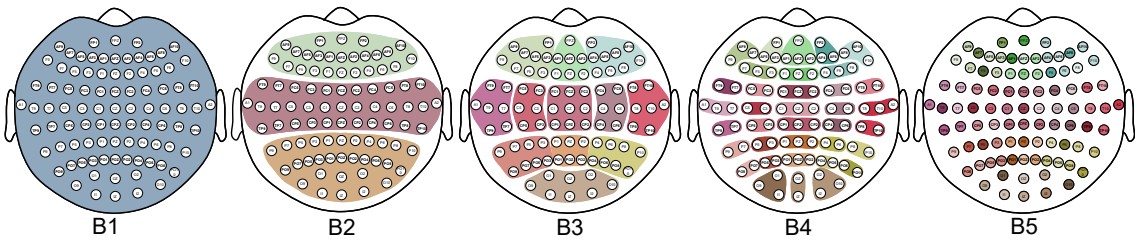

*Figure 7.* Illustration of the 5-scale Brain Topology Hierarchy (BTH).

---

**Algorithm 1** Dual-Stream Hierarchical Attention (DSHA)

---

1: **Input:** Hierarchical Features $F_{BTH} = \{F^{B_1}, \ldots, F^{B_5}\}$
2: **Modules:** Cross-Scale Block (CSB), Self-Attention (MSA)
3: *// 1. Stream Initialization*
4: $G_0 = F^{B_1}$ {Global Stream: Starts from Coarsest Level}
5: $L_0 = F^{B_5}$ {Local Stream: Starts from Finest Level}
6: *// 2. Dual-Stream Iteration (Bidirectional Interaction)*
7: **for** $k = 1$ **to** $4$ **do**
8: $\quad G_k = \text{CSB}(Q = G_{k-1}, K = F^{B_{k+1}}, V = F^{B_{k+1}})$
9: $\quad L_k = \text{CSB}(Q = L_{k-1}, K = F^{B_{5-k}}, V = F^{B_{5-k}})$
10: **end for**
11: *// 3. Intra-Stream Integration*
12: $G_{final} = \text{LN}(\text{MSA}(G_4) + G_4)$
13: $L_{final} = \text{LN}(\text{MSA}(L_4) + L_4)$
14: *// 4. Final Fusion*
15: $H_{EEG} = \text{Concat}(G_{final}, L_{final}) + \text{MeanPool}(F^{B_2 \ldots B_4})$
16: **return** Fused EEG Embedding $H_{EEG}$

---

that possesses both macro-scale coordination and micro-scale precision. The formal description of this process is provided in Algorithm 1.

## C.2. Semantic Alignment & Refinement Modules (KASP & STAR)

**1. KASP Feature Operators ($\Phi$): Mathematical Formulation and Rationale.** The KASP module utilizes a set of deterministic operators to extract robust physical features. Based on the logic defined in the main text, the detailed formulations are as follows:

- **Temporal Statistical Operator ($\phi_{stat}$):** Corresponding to the signal amplitude distribution modeling mentioned in the main text. Beyond basic central tendency (**Mean** $\mu$, **Std** $\sigma$), we explicitly introduce **Energy**, **Peak-to-Peak**, and **Kurtosis**. These supplementary statistics are crucial for capturing the signal's discreteness, transient volatility, and deviation from normality (e.g., high-amplitude artifacts or paroxysmal discharges).For a temporal channel segment $x \in \mathbb{R}^T$, the feature vector is defined as:

$$\phi_{stat}(x) = \left[ \mu, \quad \sigma, \quad \underbrace{\sum_{t=1}^{T} x_t^2}_{\text{Energy}}, \quad \underbrace{\max(x) - \min(x)}_{\text{Peak-to-Peak}}, \quad \underbrace{\frac{1}{T}\sum_{t=1}^{T}\left(\frac{x_t - \mu}{\sigma}\right)^4}_{\text{Kurtosis}} \right] \tag{19}$$

- **Spectral Feature Operator ($\phi_{spec}$):** Based on Welch's Power Spectral Density (PSD) estimation. In addition to the relative power ratios of the five standard frequency bands (Delta to Gamma), we extract **Key Rhythm Indicators** (Mean Peak Frequency and Power) to characterize the dominant oscillatory mode. Given PSD $S(f)$, for each band $b \in \{\delta, \theta, \alpha, \beta, \gamma\}$:

$$\text{Relative Power:} \quad P_{rel}(b) = \frac{\int_{f \in b} S(f)df}{\int_{0.1}^{75} S(f)df}$$
$$\text{Peak Indices:} \quad f_{peak} = \arg\max_f S(f), \quad P_{peak} = \max_f S(f) \tag{20}$$

- **Spatial Topological Operator ($\phi_{spat}$):** This operator bridges the gap between physical coordinates and semantic descriptions. It serves a dual purpose: 1) **Regional Aggregation** to summarize lobe-level neural activity; and 2) **Salient Channel Extraction (Top-K Selection)**. To prioritize the retention of the most diagnostically valuable information within a limited context window, we eschew reliance on ambiguous statistical thresholds and instead explicitly select

**Knowledge-Anchored Semantic Profiler(KASP)**

*Figure 8.* **Schematic of the Knowledge-Anchored Semantic Profiler (KASP).** The module transforms multi-task EEG inputs into rich textual descriptions through three stages: (1) individualized feature extraction to verbalize spatiotemporal characteristics; (2) structured prompt construction; and (3) LLM-based synthesis. The resulting *Semantic Profiles* contain expert-level medical priors and signal statistics to guide downstream topological feature aggregation.

the top $K$ channels exhibiting the highest statistical deviation (e.g., variance). This strategy ensures that, regardless of the absolute intensity of anomalies, the model consistently focuses on regions with the most intense or aberrant local activity (typically corresponding to lesion sites or artifact sources). Let $\mathcal{C}$ denote the set of all channels, and let $\sigma_c^2$ represent the signal variance of channel $c$. The representative channel set $C_{rep}$ is defined as the $K$ channels with the highest variance:

$$C_{rep} = \underset{c \in \mathcal{C}}{\text{Top-K}} \left( \sigma_c^2, \, K \right) \tag{21}$$

where the $\text{Top-K}$ operation returns the indices of the top $K$ channels ranked in descending order by variance.

**2. STAR: Construction Logic and Design Philosophy** The design of the Semantic Text-Aware Refiner (STAR) is rooted in the cognitive principle of **"Active Perception"**.

- **Why use "Latent Experts"?** EEG data streams are variable in length and extremely long, whereas LLMs require condensed inputs of fixed length. We initialize a fixed set of learnable vectors ($\mathcal{Q}_{lat}$) to force the model to compress the continuous signal stream into a finite set of concepts.

- **Phase 1: Expert Calibration:** Before observing the EEG data, the experts must know *what to look for*. By attending to the text generated by KASP ($H_{text}$), the latent experts are "calibrated" or "primed." For instance, if KASP describes "frontal slowing," the experts adjust their weights to pay more attention to low-frequency features in the frontal channels. This transforms generic queries into task-specific hypotheses.

- **Phase 2: Semantic Refinement:** Equipped with these hypotheses, the calibrated experts actively scan the discrete EEG token stream ($Z_q$). Unlike static pooling, this cross-attention mechanism allows the model to dynamically retrieve

signal segments that match the textual hypotheses, thereby effectively filtering out irrelevant noise and focusing on semantically aligned patterns.

The algorithmic implementation of this logic is formalized in Algorithm 2.

---

**Algorithm 2** Semantic Text-Aware Refiner (STAR)

---

1: **Input:** Text Semantics $S_{text}$ (from KASP), Discrete EEG $Z_q$
2: **Modules:** Multi-Head Cross-Attention (MHCA)
3: **Parameters:** Learnable Latent Queries $\mathcal{Q}_{lat} \in \mathbb{R}^{N_s \times E}$
4: *// Phase 1: Expert Calibration*
5: $H_{text} = \mathcal{E}_{frozen}(S_{text})$
6: $\mathcal{Q}_{calib} = \text{MHCA}(Q = \mathcal{Q}_{lat}, K = H_{text}, V = H_{text})$
7: $\mathcal{Q}_{calib} = \text{FFN}(\text{LN}(\mathcal{Q}_{calib})) + \mathcal{Q}_{calib}$
8: *// Phase 2: Semantic Refinement*
9: $O_{STAR} = \text{MHCA}(Q = \mathcal{Q}_{calib}, K = Z_q, V = Z_q)$
10: *// Phase 3: Projection*
11: $S_{sem} = \text{ProjectionHead}(O_{STAR})$
12: *// Phase 4: Orthogonality Constraint (Training Only)*
13: $M_{gram} = \frac{\mathcal{Q}_{lat}\mathcal{Q}_{lat}^T}{\|\mathcal{Q}_{lat}\|_F^2}$
14: $\mathcal{L}_{orth} = \|M_{gram} - I\|_F$
15: **return** $S_{sem}, \mathcal{L}_{orth}$

---

## C.3. Pre-training Protocols and Dynamics

**1. Stage 1: Self-Supervised Reconstruction Configuration.** We adopt a patch size of 200 (corresponding to 1 second). The model is trained using the AdamW optimizer ($\text{lr} = 5e^{-5}$) with a batch size of 128 for 50 epochs. Detailed model hyperparameters and training configurations are presented in Table 7. As shown in Figure 10(b), the training losses converge within the first 20 epochs, indicating that the model effectively captures the time-frequency characteristics of EEG data and achieves accurate reconstruction.

**2. Stage 2: Joint Autoregressive Pre-training Configuration.** We construct the hybrid sequence $\mathcal{S}$ input to the BAR model using the following format: [BOS] KASP_Profile ($S_{text}$) [SEP] STAR_Summary ($S_{sem}$) [SEP] EEG_Tokens ($S_{EEG}$) [EOS]. This explicit separation ensures that the LLM can distinguish between the static knowledge context and dynamic signal tokens.

- **LoRA Details:** We apply LoRA to the $W_q, W_k, W_v, W_o$ and MLP layers of the Qwen-2.5 backbone. Key parameters are configured as follows: Rank $r = 16$, Alpha $\alpha = 32$, and Dropout $p = 0.1$. The orthogonality weight $\lambda_{\text{orth}}$ is set to 0.1. Further detailed training hyperparameters and configurations can be found in Table 8.

Figure 10(a) illustrates the convergence of the two core NTP losses during Joint Autoregressive Pre-training. We observe that the initial Text Loss is significantly lower than the EEG Loss and exhibits a relatively smooth convergence curve, which is expected given the pre-trained nature of the LLM. However, the steady decline in EEG Loss, accompanied by the gradual increase in EEG NTP accuracy, confirms that the model successfully learns to interpret discrete EEG tokens conditioned on the KASP-STAR semantic bridge.

## C.4. Downstream Fine-tuning Configuration

**1. Multi-task Instruction Tuning Strategy.** To adapt the foundation model to specific downstream tasks while mitigating catastrophic forgetting, we employ a **Decoupled Update Strategy**:

- **Adapter Decoupling:** We freeze the pre-trained LoRA adapter or merge it into the backbone, and initialize a new, task-specific LoRA adapter for the Supervised Fine-Tuning (SFT) stage. This ensures that the general EEG-text alignment capabilities acquired during pre-training are preserved.

*Table 7.* Implementation Details of the DSHA EEG Tokenizer.

| Hyperparameters | Values |
|---|---|
| **Temporal Encoder** | |
| Input channels | {1, 16, 16} |
| Output channels | {16, 16, 16} |
| Kernel size | {15, 3, 3} |
| Stride | {8, 1, 1} |
| Padding | {7, 1, 1} |
| **Transformer Backbone** | |
| Encoder layers | 4 |
| Dual-Stream Fusion layers | 5 |
| Decoder layers | 3 |
| Hidden size | 768 |
| MLP size | 3072 |
| Attention head number | 12 |
| Codebook size | $8192 \times 128$ |
| **Training** | |
| Batch size | 128 |
| Peak learning rate | 5e-5 |
| Minimal learning rate | 1e-5 |
| Learning rate scheduler | Cosine |
| Optimizer | AdamW |
| Adam $\beta$ | (0.9, 0.999) |
| Weight decay | 1e-4 |
| Total epochs | 50 |
| Warmup epochs | 5 |
| Data overlap | None |
| Gradient clipping | None |

- **STAR Fine-tuning:** The STAR module undergoes full-parameter fine-tuning. However, to maintain the established semantic bridge, we apply a **Differential Learning Rate** strategy, setting the learning rate of STAR to $0.1\times$ that of the backbone network. This allows for subtle boundary adjustments without destroying the pre-trained alignment.

During the fine-tuning process, we compute the Cross-Entropy loss *only* on the response tokens. Gradients for system instructions, KASP profiles, expert EEG aggregates, and raw EEG history are masked out to strictly focus model updates via task-specific reasoning.

**2. Hyperparameter Configuration.** Table 9 summarizes the hyperparameters used for downstream fine-tuning in detail. Compared to pre-training, we use a smaller global Batch Size (64) to accommodate the GPU memory overhead of task-specific gradients. A cosine learning rate schedule with a warm-up ratio of 0.1 is adopted to stabilize the initial adaptation process of the new LoRA layers.

**3. Training Dynamics and Loss Curves.** Figure 10(c) illustrates the training dynamics during the SFT stage. The training loss exhibits a rapid decline within the first 2-3 epochs, significantly faster than in the pre-training stage. This rapid convergence validates the efficacy of our pre-trained representations. Concurrently, the validation perplexity (dashed line) decreases steadily alongside the training loss, indicating that the Decoupled LoRA strategy effectively maintains generalization capability and prevents overfitting to the training set.

## D. KASP Prompts and Generation Examples

This section details the prompt construction process for the Knowledge-Anchored Semantic Profiler (KASP). We employ the frozen **Qwen-2.5-7B** as the knowledge generation engine. To prevent label leakage, we adopt a **"Label-Free"** and

*Table 8.* Implementation Details for Autoregressive Continued Pre-training.

| Hyperparameters | KAST-BAR-Base | KAST-BAR-Large |
|---|---|---|
| **Brain Autoregressive Backbone** | | |
| Model size | 674M | 1872M |
| Transformer decoder layers | 24 | 28 |
| Hidden size | 896 | 1536 |
| MLP size | 4864 | 8960 |
| Attention head number | 14 | 12 |
| Key/Value heads | 2 | 2 |
| **STARefiner** | | |
| Latent queries | 16 | 32 |
| Input EEG channels | 91 | 91 |
| Refiner heads | 8 | 8 |
| **Training** | | |
| Tuning Method | LoRA | LoRA |
| LoRA Rank | 16 | 16 |
| LoRA Alpha | 32 | 32 |
| LoRA Dropout | 0.1 | 0.1 |
| Batch size | 128 | 64 |
| Peak learning rate | 5e-4 | 5e-4 |
| Minimal learning rate | 5e-5 | 5e-5 |
| Learning rate scheduler | Cosine | Cosine |
| Optimizer | AdamW | AdamW |
| Adam $\beta$ | (0.9, 0.95) | (0.9, 0.95) |
| Weight decay | 0.1 | 0.1 |
| Orthogonality Loss Weight | 0.1 | 0.1 |
| Total epochs | 20 | 20 |
| Warmup epochs | 2 | 2 |
| Data overlap | None | None |
| Gradient clipping | 1 | 1 |

**"Objectivity-Oriented"** prompting strategy, strictly prohibiting the LLM from directly outputting diagnostic conclusions.

### D.1. Feature Verbalization Logic

Before constructing the prompts, we utilize a set of feature extraction operators $\Phi$ and a verbalization mapping function to process raw EEG signals and map the results to textual descriptions. The primary extracted features include:

- **Temporal Statistical Operator** ($\phi_{stat}$)**:** Corresponds to the modeling of signal amplitude distribution described in the main text. In our implementation, we calculate the global **Mean** ($\mu$) and **Standard Deviation** ($\sigma$) across all channels. Additionally, to better capture signal discreteness and transient characteristics, we introduce **Energy**, **Peak-to-Peak Amplitude**, and **Kurtosis** as supplementary statistics.

- **Spectral Feature Operator** ($\phi_{spec}$)**:** Based on the Power Spectral Density (PSD) calculated via Welch's method. Building on the relative power ratios of the five standard frequency bands mentioned in the main text, we further extract key frequency-domain statistical indicators:

  - **Key Rhythm Indicators**: Mean Peak Frequency and Mean Peak Power.
  - **Frequency Bands**: Delta (0.5-4Hz), Theta (4-8Hz), Alpha (8-13Hz), Beta (13-30Hz), and Gamma (30-100Hz).

- **Spatial Topological Operator** ($\phi_{spat}$)**:** This operator is responsible for mapping physical coordinates to semantic descriptions. In addition to aggregating activity features from specific brain regions (e.g., frontal, occipital lobes), it

includes **Representative Channel Extraction**, which automatically identifies and reports specific electrodes exhibiting significant statistical deviations (e.g., high variance or abnormal amplitude).

### D.2. Structured Prompt Template

Figure 9 illustrates the complete prompt template used by KASP. This template dynamically concatenates the aforementioned verbalized features and compels the LLM to adopt the role of a "Data Analyst", strictly prohibiting the output of specific diagnostic labels.

### D.3. Generation Example

Table 10 presents an authentic generation result derived from a sample in the HMC (Sleep Staging) dataset. Notably, the description generated by the LLM maintains strict objectivity, validating the effectiveness of our label-free strategy.

## E. Detailed Evaluation Metrics

Given the prevalence of class imbalance in physiological signal datasets, standard accuracy is often insufficient to reflect the true performance of the model. Therefore, we employ a comprehensive set of metrics tailored to the specific nature of the tasks (binary vs. multi-class).

### E.1. Metrics for Binary Classification

For binary classification tasks with class imbalance, we define the minority class as the positive class.

**1. Balanced Accuracy (B-Acc):** B-Acc is defined as the arithmetic mean of the recall of the positive class and that of the negative class. It effectively mitigates the bias introduced by skewed class distributions.

$$\text{B-Acc} = \frac{1}{2}\left(\frac{TP}{TP + FN} + \frac{TN}{TN + FP}\right) \tag{22}$$

where $TP, TN, FP$, and $FN$ denote True Positives, True Negatives, False Positives, and False Negatives, respectively.

**2. Area Under the Receiver Operating Characteristic Curve (AUROC):** AUROC quantifies the generalization ability of the model across all classification thresholds. It is calculated as the area under the curve plotting the True Positive Rate (TPR) against the False Positive Rate (FPR). The value ranges from 0.5 to 1, with a higher value indicating better discriminative capability.

**3. Area Under the Precision-Recall Curve (AUC-PR):** In scenarios with extreme class imbalance (where positive samples are rare), AUROC may overestimate performance. AUC-PR focuses specifically on the quality of positive predictions. It is the area under the curve plotting Precision against Recall.

$$\text{Precision} = \frac{TP}{TP + FP}, \quad \text{Recall} = \frac{TP}{TP + FN} \tag{23}$$

*Table 9.* Hyperparameters for Downstream Instruction Tuning.

| Hyperparameters | KAST-BAR-Base | KAST-BAR-Large |
|---|---|---|
| Class Balancing | Yes | Yes |
| Batch size | 64 | 64 |
| Peak learning rate | 5e-4 | 1e-4 |
| Minimal learning rate | 5e-5 | 1e-5 |
| LR Scheduler | Cosine | Cosine |
| Optimizer | AdamW | AdamW |
| Adam $\beta$ | (0.9, 0.95) | (0.9, 0.95) |
| Weight decay | 0.1 | 0.1 |
| Total epochs | 10 | 5 |
| Warmup ratio | 0.1 | 0.1 |
| Gradient clipping | 1 | 1 |

**[System Instruction]**

You are a professional EEG signal data analyst. Please generate a **purely objective, technical** data report based on the provided EEG signal statistics.
**Core Principle**: The report must be divided into two parts. Part 1: General textbook-style background introduction; Part 2: Objective physical feature description.
**Strictly PROHIBITED to perform clinical diagnosis, disease judgment, or infer specific classification labels.**

**[Data Summary]**

- Sample Name: {sample_name}
- Dataset Name: {dataset_name}
- Task Logic: [System automatically maps, e.g., TUSZ → Seizure Detection]
- Channel Count: {num_channels}    Time Series Length: {num_points}

**[Verbalized Features]** ($S_{desc}$)

1. **Temporal Stats** ($\phi_{stat}$): Mean 0.12, Std 14.5, Energy 250.4...
2. **Spectral Features** ($\phi_{spec}$): Mean Peak Freq 2.5Hz, Delta Power 0.65...
3. **Spatial Features** ($\phi_{spat}$): Channel T3 (Left Temporal) shows ...

**[Analysis Requirements]**

1. **Dataset Task Description**: Describe the general experimental paradigm.
2. **Task-Related Prior Knowledge**: List relevant neuroscience background.
3. **Signal Physical Features**: Objectively describe time, frequency, and spatial features.

**[Output Format]**

Please respond strictly in the following JSON format:
```
{
    "Dataset Task Description": "General experimental paradigm introduction...",
    "Task Related Prior Knowledge": "General medical/neuroscience background",
    "Signal Physical Features": "Purely objective description...",
    "Spatial Distribution Features": "Prominent brain regions",
    "Data Quality Notes": "Outlier channels or noise notes",
    "Feature Summary": "Morphology summary (NO diagnosis)"
}
```

*Figure 9.* The structured prompt template used in KASP. Different sections are highlighted with distinct colors to indicate their semantic roles (e.g., orange for instructions, blue for data context, green for features).

## E.2. Metrics for Multi-class Classification

For multi-class tasks such as sleep staging (HMC) and emotion recognition (SEED), we utilize the following metrics.

**1. Balanced Accuracy (B-Acc):** In the multi-class setting, B-Acc is defined as the macro-average of the recall scores for all classes.

$$\text{B-Acc} = \frac{1}{C} \sum_{i=1}^{C} \text{Recall}_i \tag{24}$$

where $C$ is the total number of classes, and $\text{Recall}_i$ is the recall for class $i$.

**2. Cohen's Kappa Coefficient ($\kappa$):** Cohen's Kappa measures the agreement between the model's predictions and the ground truth labels, correcting for agreement occurring by chance.

$$\kappa = \frac{p_o - p_e}{1 - p_e} \tag{25}$$

where $p_o$ represents the observed agreement (accuracy), and $p_e$ is the expected agreement by chance.

**3. Weighted F1-Score (F1-W):** To balance precision and recall while accounting for the support of each class, we employ the Weighted F1-Score. It is calculated as the weighted sum of per-class F1-scores, where the weight $w_i$ corresponds to the

*Table 10.* Example of KASP-generated Semantic Profile (Based on HMC Dataset, Sample SN022)

| JSON Key | Generated Content (Value) |
|---|---|
| `Dataset Task Description` | This is a sleep staging task designed to investigate sleep patterns and associated EEG characteristics across different stages. |
| `Task Related Prior Knowledge` | The brain exhibits distinct brainwave characteristics at different sleep stages, such as Delta waves (typically dominant during deep sleep in NREM stages) and Theta waves (common in light sleep stages). |
| `Signal Physical Features` | The overall channel mean is -0.00023, with a standard deviation of 40.85, energy of 1668.67, and peak value of 1769.40. Channel C4 exhibits the maximum power ratio in the Delta band, with a peak power of 524.68 and relative power of 49.81%; Channel C3 has a peak power of 478.24 and relative power of 48.51%; F4 shows a relative power of 47.38%; O2 shows a relative power of 44.22%. |
| `Spatial Distribution Features` | Channels C4 and C3 exhibit high power ratios in the Delta band. |
| `Data Quality Notes` | No obvious outlier channels or noise interference observed; however, the overall energy and peak values of the recording are high, necessitating further inspection to assess potential artifactual effects. |
| `Feature Summary` | The sample is primarily characterized by low-frequency activity in the frontal brain region, particularly within the Delta band. |

proportion of samples in class $i$.

$$\text{F1}_i = \frac{2 \times \text{Precision}_i \times \text{Recall}_i}{\text{Precision}_i + \text{Recall}_i} \tag{26}$$

$$\text{F1-W} = \sum_{i=1}^{C} w_i \cdot \text{F1}_i, \quad w_i = \frac{N_i}{N_{total}} \tag{27}$$

## F. Additional Experimental Results

We provide a comprehensive performance analysis of KAST-BAR across six diverse downstream datasets, comparing it against leading single-task specialists and multi-task generalist models. Detailed numerical results are presented in Table 11, Table 12, and Table 13.

*Table 11.* Performance comparison on Workload and TUSL datasets. "General Model" indicates if the model is pre-trained for general representations, and "Multi-Task" indicates if it's designed for or evaluated on multiple tasks. Best results are **bold** for multi-task methods and underlined for single-task methods.

| Methods | Model Parameter | General Model | Multi-Task | Workload | | | TUSL | | |
|---|---|---|---|---|---|---|---|---|---|
| | | | | B-Acc | AUC-PR | AUROC | B-Acc | Kappa | F1-W |
| EEGNet (Lawhern et al., 2018) | - | ✗ | ✗ | 60.9±2.6 | 58.2±2.0 | 62.4±1.5 | 57.4±4.2 | 49.8±4.5 | 52.9±6.8 |
| SPaRCNet (Jing et al., 2023) | 0.79M | ✗ | ✗ | 59.8±0.7 | 66.4±3.1 | 67.2±1.7 | 56.9±2.5 | 49.2±4.0 | 58.4±4.9 |
| ST-Transformer (Song et al., 2021) | 3.5M | ✗ | ✗ | 61.0±0.6 | 57.2±0.7 | 63.8±0.8 | 49.3±5.6 | 30.2±10.0 | 41.2±5.9 |
| BIOT (Yang et al., 2023) | 3.2M | ✓ | ✗ | 66.6±1.1 | 71.9±7.2 | 73.4±5.4 | 57.6±3.0 | 20.1±2.1 | 23.9±0.4 |
| LaBraM-Base (Jiang et al., 2024) | 5.8M | ✓ | ✗ | 66.1±2.0 | 71.7±2.3 | 72.7±1.7 | 76.3±2.3 | 64.1±3.0 | 76.1±2.1 |
| CBraMod (Wang et al., 2025a) | 4.0M | ✓ | ✗ | 65.4±2.6 | 70.4±1.3 | 70.1±2.3 | 73.9±3.2 | 61.5±5.5 | 74.5±3.6 |
| CSBrain (Zhou et al., 2025) | 4.9M | ✓ | ✗ | —— | —— | —— | 85.7±2.4 | 78.3±2.7 | 85.7±1.8 |
| EEGPT (Wang et al., 2024) | 25M | ✓ | ✓ | 63.0±1.8 | 67.9±0.9 | 69.3±1.1 | 72.9±1.4 | 59.7±2.1 | 72.3±1.5 |
| NeuroLM-XL (Jiang et al., 2025) | 1.7B | ✓ | ✓ | 63.5±4.4 | 58.9±4.2 | 61.3±7.6 | 68.5±3.0 | 52.0±4.6 | 68.4±3.0 |
| THD-BAR-Huge (Yang et al., 2025) | 1.6B | ✓ | ✓ | 67.1±5.8 | 60.1±3.7 | 63.6±6.2 | 69.2±2.3 | 53.4±3.3 | 67.1±1.7 |
| KAST-BAR-Base(Ours) | 0.8B | ✓ | ✓ | 68.6±2.1 | 64.2±2.9 | 67.3±1.6 | 73.7±3.3 | 63.1±2.7 | 74.8±2.9 |
| KAST-BAR-Large(Ours) | 2.2B | ✓ | ✓ | **71.2±1.7** | **69.5±2.0** | **73.1±3.5** | **77.4±3.9** | **65.4±3.0** | **76.6±2.4** |

*Table 12.* Performance comparison on TUAB and TUEV datasets. "General Model" indicates if the model is pre-trained for general representations, and "Multi-Task" indicates if it's designed for or evaluated on multiple tasks. Best results are **bold** for multi-task methods and underlined for single-task methods.

| Methods | Model Parameter | General Model | Multi-Task | TUAB | | | TUEV | | |
|---|---|---|---|---|---|---|---|---|---|
| | | | | B-Acc | AUC-PR | AUROC | B-Acc | Kappa | F1-W |
| EEGNet (Lawhern et al., 2018) | - | ✗ | ✗ | 77.1±0.6 | 82.3±0.3 | 85.0±0.3 | 39.8±1.1 | 34.9±1.1 | 63.8±2.1 |
| SPaRCNet (Jing et al., 2023) | 0.79M | ✗ | ✗ | 77.5±1.0 | 83.1±0.7 | 86.3±0.6 | 43.2±3.2 | 43.8±2.2 | 68.1±1.7 |
| ST-Transformer (Song et al., 2021) | 3.5M | ✗ | ✗ | 79.3±0.2 | 85.4±0.6 | 86.9±0.2 | 39.6±1.8 | 38.3±2.3 | 69.1±2.1 |
| BIOT (Yang et al., 2023) | 3.2M | ✓ | ✗ | 79.6±0.6 | 87.9±0.2 | 88.2±0.4 | 52.8±2.2 | 52.7±2.5 | 74.9±0.8 |
| LaBraM-Base (Jiang et al., 2024) | 5.8M | ✓ | ✗ | 81.4±0.2 | 89.7±0.1 | 90.2±0.1 | 64.1±0.7 | 66.4±1.0 | 83.1±0.5 |
| CBraMod (Wang et al., 2025a) | 4.0M | ✓ | ✗ | 78.9±0.3 | 86.4±0.6 | 86.1±0.6 | 66.7±1.1 | 67.7±1.0 | 83.4±0.6 |
| CSBrain (Zhou et al., 2025) | 4.9M | ✓ | ✗ | 81.7±0.4 | 90.1±0.7 | 89.6±0.5 | 69.0±0.6 | 68.3±0.5 | 83.3±0.6 |
| EEGPT (Wang et al., 2024) | 25M | ✓ | ✓ | 79.2±0.4 | 74.6±0.5 | 86.6±0.7 | 62.3±1.1 | 63.5±1.3 | 81.9±0.6 |
| NeuroLM-XL (Jiang et al., 2025) | 1.7B | ✓ | ✓ | 79.7±0.9 | 72.2±0.8 | 78.8±1.9 | 46.8±3.6 | 45.7±5.0 | 73.6±2.2 |
| THD-BAR-Huge (Yang et al., 2025) | 1.6B | ✓ | ✓ | 82.2±0.4 | 84.8±0.7 | 88.6±1.2 | 65.3±0.5 | 64.4±2.3 | 82.1±1.4 |
| KAST-BAR-Base(Ours) | 0.8B | ✓ | ✓ | 81.3±0.6 | 85.6±0.5 | 88.2±1.4 | 66.1±0.9 | 64.7±1.8 | 82.8±1.5 |
| KAST-BAR-Large(Ours) | 2.2B | ✓ | ✓ | **83.2±1.1** | **87.1±1.3** | **90.7±2.1** | **70.8±1.4** | **69.6±2.2** | **85.5±1.1** |

*Table 13.* Performance comparison on SEED and HMC datasets. "General Model" indicates if the model is pre-trained for general representations, and "Multi-Task" indicates if it's designed for or evaluated on multiple tasks. Best results are **bold** for multi-task methods and underlined for single-task methods.

| Methods | Model Parameter | General Model | Multi-Task | SEED | | | HMC | | |
|---|---|---|---|---|---|---|---|---|---|
| | | | | B-Acc | Kappa | F1-W | B-Acc | Kappa | F1-W |
| EEGNet (Lawhern et al., 2018) | - | ✗ | ✗ | 62.5±3.5 | 44.7±2.3 | 61.8±2.7 | 62.2±2.1 | 56.1±1.9 | 62.8±1.4 |
| SPaRCNet (Jing et al., 2023) | 0.79M | ✗ | ✗ | 56.0±2.4 | 34.6±3.7 | 55.9±3.0 | 55.4±3.7 | 47.9±3.0 | 57.0±4.3 |
| ST-Transformer (Song et al., 2021) | 3.5M | ✗ | ✗ | 56.4±1.8 | 37.4±1.8 | 56.3±1.4 | 59.5±5.4 | 50.0±2.0 | 61.5±3.3 |
| BIOT (Yang et al., 2023) | 3.2M | ✓ | ✗ | 71.0±0.2 | 56.8±0.5 | 71.3±0.3 | 68.6±0.4 | 63.0±1.1 | 70.9±1.5 |
| LaBraM-Base (Jiang et al., 2024) | 5.8M | ✓ | ✗ | 73.2±0.2 | 59.9±0.3 | 73.5±0.2 | 72.9±1.0 | 68.1±0.7 | 75.5±0.2 |
| CBraMod (Wang et al., 2025a) | 4.0M | ✓ | ✗ | 72.7±0.7 | 57.6±0.2 | 73.0±0.4 | 72.7±0.4 | 66.9±1.0 | 74.0±0.9 |
| CSBrain (Zhou et al., 2025) | 4.9M | ✓ | ✗ | 73.0±0.4 | 59.3±0.3 | 73.0±0.4 | 73.5±0.5 | 68.2±0.5 | 75.1±0.4 |
| EEGPT (Wang et al., 2024) | 25M | ✓ | ✓ | 71.2±0.2 | 57.3±0.5 | 71.0±0.4 | 70.3±0.8 | 65.8±0.6 | 73.2±0.4 |
| NeuroLM-XL (Jiang et al., 2025) | 1.7B | ✓ | ✓ | 60.3±0.1 | 40.8±0.4 | 60.6±0.3 | 57.6±10.8 | 48.0±14.7 | 58.8±12.9 |
| THD-BAR-Huge (Yang et al., 2025) | 1.6B | ✓ | ✓ | 73.9±0.3 | 58.9±0.3 | **72.7±0.2** | 68.4±4.3 | 63.5±1.3 | 71.8±1.1 |
| KAST-BAR-Base(Ours) | 0.8B | ✓ | ✓ | 72.7±0.4 | 58.1±0.5 | 71.9±0.9 | 72.5±2.1 | 69.1±0.9 | 74.9±1.0 |
| KAST-BAR-Large(Ours) | 2.2B | ✓ | ✓ | **74.3±0.2** | **60.6±0.8** | 72.2±1.2 | **73.9±1.7** | **70.5±1.4** | **76.2±1.5** |

## F.1. Comparison with Multi-Task Baselines

As shown in the tables, **KAST-BAR-Large** consistently outperforms existing multi-task foundation models (including NeuroLM, EEGPT, and THD-BAR) across all evaluation benchmarks.

On complex clinical tasks such as **TUEV** (Event Detection), KAST-BAR-Large achieves a Balanced Accuracy (B-Acc) of **70.8%**, significantly surpassing THD-BAR-Huge (65.3%) and NeuroLM-XL (46.8%); meanwhile, consistent improvements in Kappa and F1-W metrics further corroborate its superior classification capabilities. This indicates that our **KASP** and **STAR** modules effectively bridge the semantic gap, enabling the model to precisely distinguish complex clinical events that simple instruction-following models fail to capture by injecting fine-grained medical knowledge embeddings.

In the **Workload** (Cognitive Load) estimation task, KAST-BAR reaches a B-Acc of **71.2%**, leading the strongest previous multi-task competitor (THD-BAR, 67.1%) by a substantial margin of 4.1%. This proves that our **DSHA encoder** possesses strong robustness in handling drastic cross-subject variations in cognitive states through explicit bidirectional topological information interaction.

In the **SEED** (Emotion Recognition) task, the improvement of KAST-BAR-Large over THD-BAR-Huge is relatively moderate (+0.4%). We attribute this to the fact that emotion recognition tasks rely heavily on the **spatial asymmetry features** of brain hemispheres. The hierarchical topological constraints introduced by THD-BAR have already captured these static spatial patterns to some extent, resulting in a high baseline. However, KAST-BAR still maintains superior performance,

*Table 14.* Supplementary ablations. **Part I** evaluates the impact of substituting the LLM backbone. **Part II** demonstrates the effect of the orthogonality loss ($\mathcal{L}_{orth}$) on expert diversity.

| Model Variant | Configuration | Measured $\mathcal{L}_{orth}$ | TUEV (B-Acc) | HMC (B-Acc) |
|---|---|---|---|---|
| *Part I: Impact of LLM Backbone* | | | | |
| THD-BAR-Huge | GPT-2 | - | 65.3 | 68.4 |
| THD-BAR-Huge | Qwen2.5-1.5B | - | 66.5 | 70.1 |
| KAST-BAR-Large | Qwen2.5-1.5B | 0.03 | **70.8** | **73.9** |
| *Part II: Impact of Orthogonality Loss ($\mathcal{L}_{orth}$)* | | | | |
| KAST-BAR-Large | w/o $\mathcal{L}_{orth}$ | 0.81 | 68.3 | 72.1 |
| KAST-BAR-Large | Full | **0.03** | **70.8** | **73.9** |

demonstrating that even in tasks with relatively simple semantic descriptions, the **dynamic** topological refinement provided by DSHA can further mine subtle cortical activity differences ignored by static models.

### F.2. Comparison with Single-Task Specialists

It is worth noting that although KAST-BAR is a generalist model dealing with heterogeneous distributions, it matches or even exceeds the performance of single-task models fine-tuned specifically for certain datasets (such as LaBraM, CSBrain, and BIOT) on multiple tasks, though there remains a gap compared to single-task specialists on individual datasets.

**Superiority on Heterogeneous Tasks:** On **TUAB** (Abnormal Detection) and **HMC** (Sleep Staging), KAST-BAR-Large achieves superior performance (83.2% and 73.9% B-Acc, respectively), surpassing the highly optimized single-task model CSBrain (81.7% and 73.5%). This suggests that "universal knowledge" learned from massive pre-training can synergize with task-specific prompts to generate superior representations. **Analysis of the TUSL Exception:** We observe that on the **TUSL** (Slowing Detection) dataset, the single-task model **CSBrain** outperforms KAST-BAR-Large (85.7% vs. 77.4% B-Acc). We attribute this to two primary factors:

1. **Advantage of Cross-scale Feature Capture:** CSBrain introduces a Cross-scale Spatiotemporal Tokenization (CST) mechanism, designed to explicitly aggregate local high-frequency transients and global low-frequency rhythms. For specific spectral anomalies in the TUSL task (such as slowing waves), this multi-scale inductive bias is more effective at precisely locking onto local pathological features than KAST-BAR's strategy, which focuses on global topological connectivity.

2. **Negative Transfer in Multi-Task Learning:** TUSL represents a highly specific clinical anomaly detection task, with a data distribution vastly different from tasks like emotion recognition or cognitive workload. In unified multi-task modeling, forcibly optimizing these semantically conflicting tasks simultaneously may introduce gradient interference, leading to slight "negative transfer" for KAST-BAR on such narrow-distribution tasks. However, KAST-BAR still significantly outperforms other multi-task baselines (e.g., THD-BAR-Huge: 69.2%), proving its overall robustness.

### F.3. Scaling Analysis

The results also highlight the efficacy of model scaling. **KAST-BAR-Large** (2.2B parameters) consistently outperforms **KAST-BAR-Base** (0.8B) across all metrics. For instance, on TUEV, the Large model improves B-Acc by 4.7% (66.1% $\rightarrow$ 70.8%) and F1-W by 2.7% compared to the Base model. This validates that increasing model capacity, combined with our dynamic topology-aware architecture, leads to stronger generalization and semantic reasoning capabilities.

### F.4. Additional Ablation Studies

**Impact of the Orthogonality Loss ($\mathcal{L}_{orth}$):** We further ablated the orthogonality penalty ($\mathcal{L}_{orth}$) to evaluate its role in preventing query redundancy. **As shown in Table 14 (Part II)**, without $\mathcal{L}_{orth}$, the measured orthogonality value surges to 0.81, indicating that latent experts collapse into redundant representations. Activating the penalty drops this metric to 0.03, ensuring expert diversity and yielding accuracy gains of 2.5% on TUEV and 1.8% on HMC.

**Architectural Innovation vs. Model Scaling:** To isolate the performance gains attributed to the LLM backbone, **as detailed in Table 14 (Part I)**, we upgraded the THD-BAR baseline from GPT-2 to Qwen2.5-1.5B. This upgrade yielded only minor

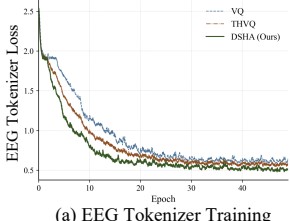 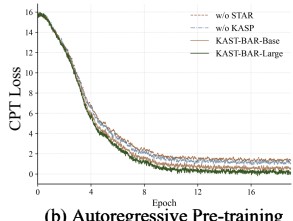 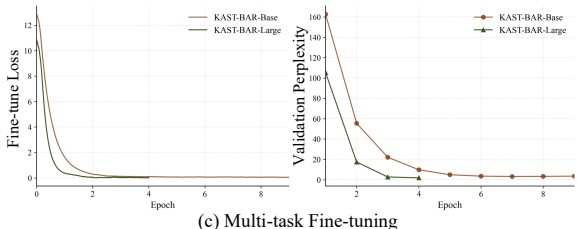

| (a) EEG Tokenizer Training | (b) Autoregressive Pre-training | (c) Multi-task Fine-tuning |

*Figure 10.* Training and fine-tuning dynamics of KAST-BAR. (a) EEG tokenizer reconstruction loss, highlighting the superiority of DSHA over VQ and THVQ; (b) Autoregressive pre-training (CPT) loss comparison; (c) Multi-task fine-tuning metrics, where the KAST-BAR-Large model demonstrates faster adaptation in Fine-tune Loss (left) and lower Validation Perplexity (right) compared to the KAST-BAR-Base model.

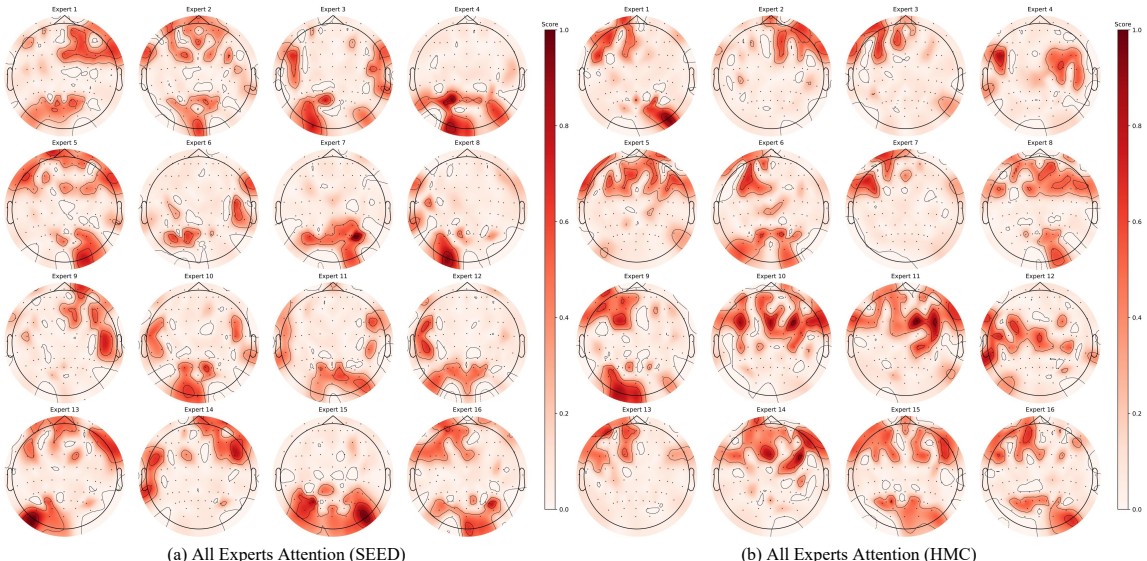

| (a) All Experts Attention (SEED) | (b) All Experts Attention (HMC) |

*Figure 11.* Visualization of spatial attention weights for STAR experts on (a) SEED (Emotion Recognition) and (b) HMC (Sleep Staging). The distinct patterns—focusing on temporal-occipital areas for emotion versus frontal-central areas for sleep—verify the model's capability to capture task-specific topological dependencies.

improvements (+1.2% on TUEV and +1.7% on HMC), which still falls significantly short of KAST-BAR's final performance. This confirms that KAST-BAR's superiority fundamentally stems from our proposed architectural innovations-DSHA, KASP, and STAR—rather than merely scaling the foundation LLM.

# G. Visualization and Qualitative Analysis

## G.1. Training Dynamics

To comprehensively evaluate the stability and internal mechanisms of the proposed framework, we analyzed the training curves and visualized the full spectrum of latent expert attention of the KAST-BAR-Base model across two datasets. Figure 10 illustrates the learning dynamics during different training phases. In the EEG Tokenizer training stage as Figure 10(a), our proposed DSHA demonstrates significantly superior performance compared to traditional VQ and THVQ methods. The loss curve for DSHA exhibits a steeper initial descent and converges to a lower bound, confirming that explicitly modeling the interaction between local neural dynamics and global spatiotemporal contexts enables the tokenizer to capture the non-Euclidean manifold of brain signals with higher fidelity.

Regarding scalability and generalization capabilities, the autoregressive pre-training curves in Figure 10(b) reveal stable convergence in CTP loss, indicating effective learning dynamics. Notably, KAST-BAR-Large consistently outperforms the Base version, exhibiting a significantly steeper loss reduction trajectory and faster convergence rate. This indicates that,

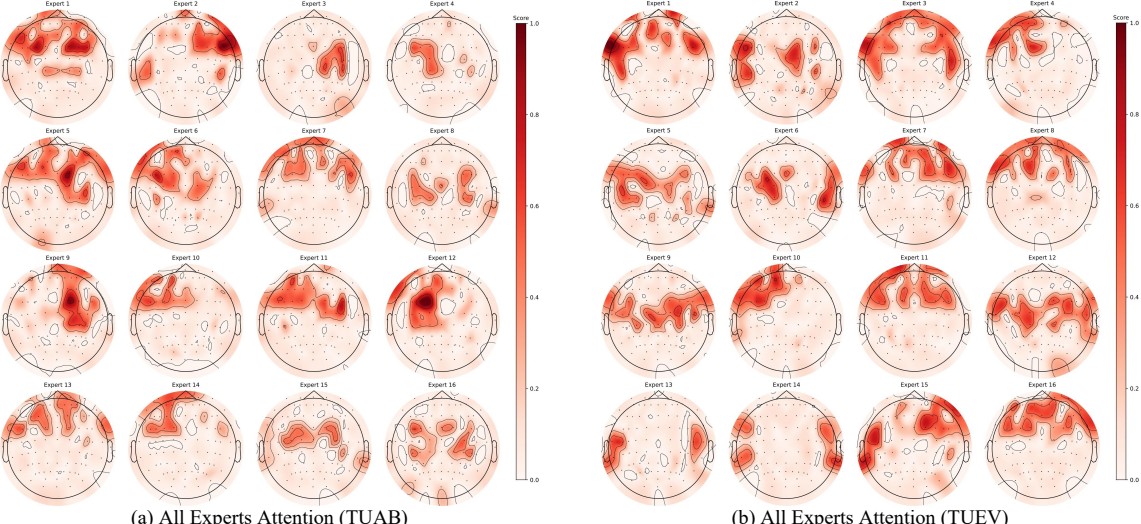

(a) All Experts Attention (TUAB)      (b) All Experts Attention (TUEV)

*Figure 12.* Visualization of spatial attention weights on (a) TUAB (Abnormal Detection) and (b) TUEV (Epilepsy Events). The experts demonstrate distributed attention patterns corresponding to pathological features and transient epileptic events across different brain regions.

rather than being hindered by larger capacity, the scaled-up backbone—synergized with our semantic alignment—more efficiently captures complex physiological patterns. This advantage directly translates to superior generalization, facilitating rapid downstream adaptation. In the multi-task fine-tuning stage as Figure 10(c), the validation perplexity and loss of the Large model drop rapidly. Both the convergence and descent rates are significantly higher than those of the Base model. Leveraging richer semantic-physical priors accumulated during pre-training and stronger generalization capabilities, the Large model efficiently adapts to downstream tasks with fewer iteration steps. These results confirm that scaling up model capacity significantly enhances the ability to absorb medical priors and signal patterns, thereby facilitating robust adaptation in downstream tasks.

### G.2. Visualization of Knowledge-Driven Topological Refinement

To elucidate the interpretability of our "Topological Encoding - Knowledge Anchoring - Guided Refinement" paradigm, we visualize the spatial attention distributions of the 16 latent expert queries generated by the **Semantic Text-Aware Refiner (STAR)** of the **KAST-BAR-Base** model. As illustrated in Figures 11, 12, 13, these heat maps represent the physical manifestation of how the **Knowledge-Anchored Semantic Profiler (KASP)** actively orchestrates the experts to aggregate EEG features from the non-Euclidean 3D manifold. Rather than processing information uniformly, the experts act as dynamic probes, actively searching for clinically relevant patterns defined by task-specific medical insights synthesized by KASP.

The effectiveness of this knowledge-driven orchestration is first evident in the distinct topological configurations observed between emotional and sleep tasks. In the SEED emotion recognition task (Figure 11(a)), guided by KASP's semantic synthesis of "visual stimulation" and "emotional arousal," the experts spontaneously form a functional cluster. Specific queries (e.g., Experts 2, 3, 15) heavily attend to the *Occipital Lobe* (visual processing) and *Temporal Lobe* (emotional regulation), validating that STAR successfully decouples complex signals by anchoring on the "visual-emotional" semantic priors. In sharp contrast, Figure 11(b) reveals a drastic topological shift on the HMC sleep staging dataset. Upon receiving context regarding sleep micro-events, the latent experts (e.g., Experts 2, 10, 12) realign their attention towards the *Frontal* and *Central* regions. This spatial distribution perfectly coincides with the physiological generation sites of key biomarkers such as K-complexes and Sleep Spindles, demonstrating that the dynamic refinement mechanism allows the model to adaptively select relevant physiological features conditioned on textual priors.

Furthermore, this "Knowledge-Anchored" mechanism demonstrates robust generalization across cognitive and pathological domains. In the Workload assessment task (Figure 13(b)), the experts reconstruct the Central Executive Network (CEN) essential for working memory processing. Conversely, in the TUSL pathology detection task (Figure 13(a)), the attention

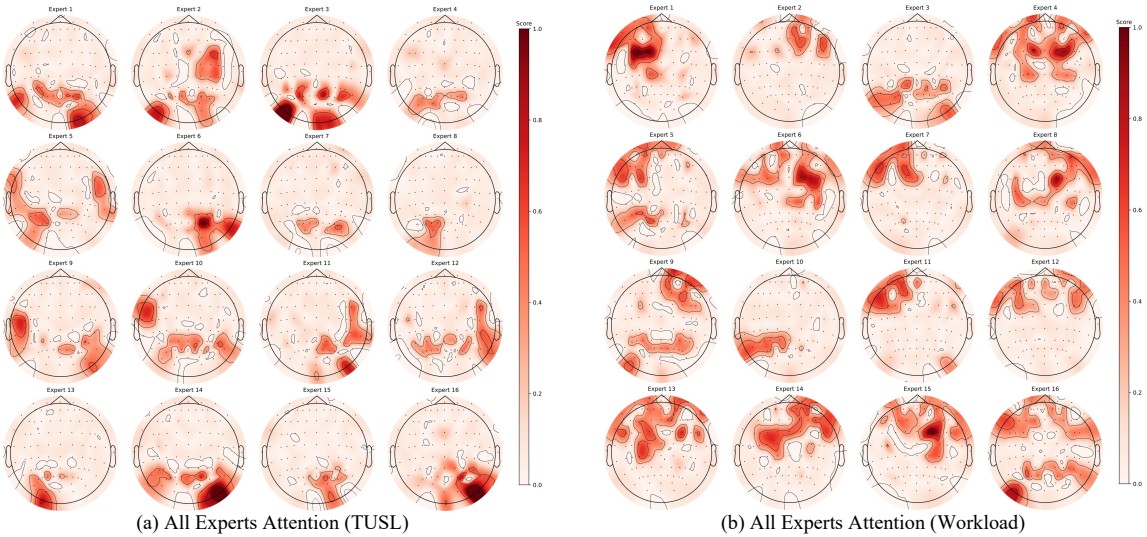

(a) All Experts Attention (TUSL)     (b) All Experts Attention (Workload)

*Figure 13.* Visualization of spatial attention weights on (a) TUSL (Slowing Event) and (b) Workload Assessment. The heatmaps highlight the topological regions most relevant to detecting background slowing and estimating cognitive load levels.

becomes focal and lateralized to pinpoint pathological slowing events. These distinct topological reconfigurations strongly evidence that KAST-BAR goes beyond implicit alignment. By synergizing knowledge-driven semantic reasoning with dynamic topological perception, the model achieves interpretable, expert-level signal understanding without relying on explicit supervision or static geometric constraints.

## H. Limitations and Future Work

Despite KAST-BAR achieving superior performance on most semantics-dominated tasks, limitations persist in specific scenarios. First, the model's performance on the TUSL (Slowing detection) task is slightly inferior to specialized models, revealing a **Feature Granularity Mismatch** in the architecture: the DSHA encoder tends to extract high-level global topological features, causing microscopic frequency details to potentially be smoothed out during quantization. Second, as a model built upon a billion-parameter LLM backbone, KAST-BAR incurs significantly higher computational complexity than traditional lightweight models. Furthermore, the autoregressive generation of the KASP module introduces inference latency, restricting its direct deployment in millisecond-level real-time BCI systems. Additionally, reliance on LLM-based semantic priors introduces potential "hallucination" risks; in extremely rare pathological scenarios, medical knowledge provided by general-purpose language models may lack precision, and the model risks inheriting statistical biases from training data.

To address these challenges, our future work will focus on the following directions. To resolve the granularity trade-off, we plan to introduce an **adaptive multi-granularity Tokenizer** or hybrid loss functions, allowing the model to dynamically adjust temporal resolution according to task demands, thereby enhancing the reconstruction capability of local waveform details while maintaining semantic generalization. Addressing real-time deployment needs, we will explore **Knowledge Distillation** techniques, utilizing KAST-BAR as a teacher model to transfer its universal topological insights to lightweight student models suitable for edge computing devices. Meanwhile, we plan to integrate **structured Medical Knowledge Graphs** into KASP to constrain text generation, ensuring the accuracy of clinical priors. Finally, we will prioritize neuroethics, ensuring the fairness and safety of the model as an assistive diagnostic tool across diverse populations.

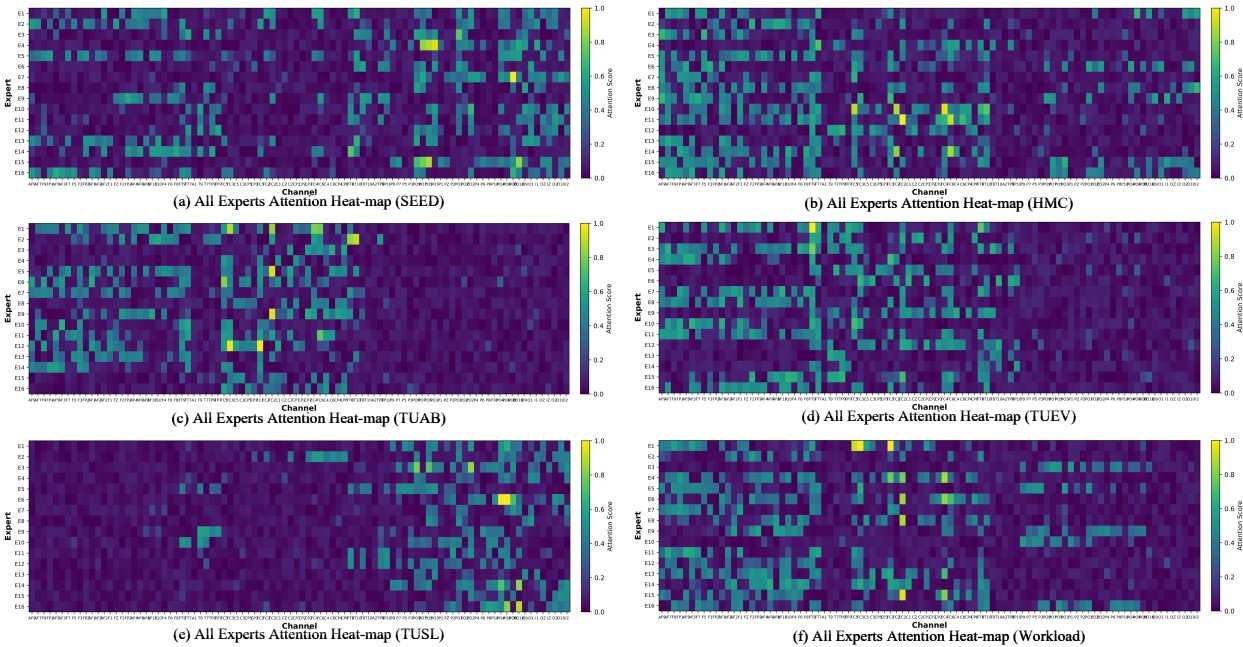

*Figure 14.* Visualization of attention heatmaps for 16 latent experts across six datasets: (a) SEED, (b) HMC, (c) TUAB, (d) TUEV, (e) TUSL, and (f) Workload. The y-axis represents the latent experts, and the x-axis represents the EEG channels, with color intensity indicating the magnitude of attention weights.

