# OpenReview forum: "KAST-BAR: Knowledge-Anchored Semantically-Dynamic Topology Brain Autoregressive Modeling for Universal Neural Interpretation"
_ICML.cc/2026/Conference — ICML 2026 regular_

### Official Review · Reviewer_rP81 · 2026-03-10

**Soundness:** 3
**Presentation:** 3
**Significance:** 3
**Originality:** 2
**Overall Recommendation:** 4
**Confidence:** 4

**Summary:**

This paper proposes an EEG foundation model called KAST-BAR for dynamic alignment of physiological representations by the multi-level brain topology with an expert-level semantic space. The core architecture proposed in this paper consists of three components. The first component, DSHA, models brain topological characteristics from EEG data through dual-stream hierarchical attention. The second component, KASP, leverages domain knowledge and basic EEG signal properties to generate instance-level textual profiles. The third component, STAR, ultimately aggregates EEG features and textual semantic information via latent expert query vectors. The entire model is trained through a three-stage process: the first two stages involve pre-training on 21 datasets, while the third stage consists of fine-tuning on six distinct EEG downstream tasks, ultimately achieving outstanding performance.

**Compliance With Llm Reviewing Policy:**

Affirmed.

**Final Justification:**

We have already provided a detailed explanation of the paper's strengths and weaknesses during the review phase. The authors addressed our main concerns during the discussion phase, and we will maintain our score.

**Key Questions For Authors:**

1. Could the authors incorporate the EEG reconstruction process and annotate some of the key variables from the equations within the framework diagram to provide readers with a clearer and more intuitive understanding?

2. Could the authors provide a detailed description of the spatial attention acquisition process of the STAR experts, so that readers can understand which module's parameters were visualized in the paper?

3. Could the authors clarify through evaluation whether the adoption of Qwen2.5 indeed leads to superior performance in certain modules, or alternatively, demonstrate that Qwen2.5 does not yield significant improvements over GPT-2 in terms of model performance?

**Limitations:**

Yes.

**Strengths And Weaknesses:**

Strengths

S1. The paper presents a substantial workload and makes notable contributions to the engineering application of large EEG models, while also achieving excellent performance across multiple tasks.

S2. The paper is well-organized and logically coherent, with clear introductions to its core modules and formulations of equations that are easy to follow. Additionally, there are almost no errors in expression throughout the manuscript.

S3. The latent expert queries obtained in the paper are visualized, which renders the model more interpretable compared to some large-scale EEG pre-trained models.

Weaknesses

W1. The core modules of the paper are limited in terms of novelty, representing primarily refinements in architectural design and the integration of EEG with large models using existing techniques. Consequently, the work lacks original design and theoretical depth.

W2. The paper contains a substantial number of mathematical formulations. If key output variables could be annotated within the framework diagram, it would further enhance the clarity and presentation of the paper.

W3. The reconstruction process of EEG signals is not clearly illustrated in the overall framework diagram. Moreover, although the input to the DSHA module is raw time-domain EEG signals, the decoding process employs vector quantization to simultaneously reconstruct both frequency-domain and time-domain EEG signals. The motivation behind this design choice is insufficiently discussed, and its specific contribution to the overall model performance remains unclear.

W4. Regarding the visualization of the STAR experts, the process for obtaining spatial attention is not clearly explained, making it difficult to understand what specific part of the model the visualization actually corresponds to.

W5. The base large model adopted in this paper is Qwen2.5. Compared to prior work that utilized the GPT-2 series, the extent to which this modification contributes to the model's performance gains remains unclear.

---

> ### Author Rebuttal · Authors · 2026-03-31
>
> We sincerely thank Reviewer rP81 for their constructive feedback and for recognizing our substantial workload, excellent multi-task performance, logical coherence, and model interpretability. Detailed responses addressing your concerns are provided below, and we will incorporate these clarifications into the final version.
>
> **[Q1 / W2 / W3] Framework diagram annotations & VQ reconstruction motivation**
>
> We will include a comprehensive EEG reconstruction flowchart in the final version, annotated with key variables like $H_{EEG}$ and $Z_q$, to enhance intuitive understanding. Our motivation for employing Vector Quantization (VQ) for joint time-frequency reconstruction is twofold:
>
> 1. **Signal Characteristics**: EEG is a highly complex, non-stationary signal. While the input consists of raw time-domain signals, the VQ decoder simultaneously reconstructs both the time-domain signals and its DFT-extracted frequency-domain signals (Jiang et al., ICLR 2024). Time-domain reconstruction preserves transient micro-events (e.g., epileptic spikes), while frequency-domain reconstruction captures macro-rhythmic oscillations (e.g., sleep spindles).
> 2. **Baseline Consistency:** VQ-based joint reconstruction is a standard paradigm widely adopted by EEG foundation models , such as LaBraM (Jiang et al., ICLR 2024), NeuroLM (Jiang et al., ICLR 2025), and THD-BAR (Yang et al., NeurIPS 2025). Following this paradigm ensures aligned pre-training objectives, enabling a fair validation of our DSHA module's superiority.
>
> **[Q2 / W4] Visualization of STAR experts' spatial attention**
>
> The visualizations in the manuscript (Figures 5-6 and 11-14) show the cross-attention weight matrices from the second stage of the STAR module (EEG Refinement). We mapped these channel-wise weights to the brain topology, illustrating how specific text semantics drive the latent experts to adaptively focus on distinct brain topological regions.
>
> Specifically, the learnable latent queries $\mathcal{Q}\_{lat}$ are first calibrated by sample-specific text semantics $H_{text}$ to yield text-calibrated queries $\mathcal{Q}_{calib}$, which then serve as the Query to dynamically aggregate the discrete EEG topological features $Z_q$ (acting as Key and Value). The visualized attention weights are extracted directly from this specific MHCA operation, which corresponds to **Line 9 in Algorithm 2**:
>
>  $O_{STAR}=MHCA(Q=\mathcal{Q}_{calib}, K=Z_q, V=Z_q)$.
>
> We will further clarify this extraction process in the final version.
>
> **[Q3 / W5] Contribution of the Qwen2.5 backbone vs. GPT-2**
>
> We first clarify a practical constraint: testing KAST-BAR directly with a GPT-2 backbone is unfeasible because the combined sequence length of our Semantic Profiles and EEG tokens exceeds legacy GPT-2's maximum context capacity. Thus, to explore the impact of LLM backbone, we instead upgraded the baseline THD-BAR model from GPT-2 to Qwen2.5 (1.5B) and evaluated it on the TUEV and HMC datasets:
>
> | Model | Backbone | TUEV (B-Acc) | HMC (B-Acc) |
> | --- | --- | --- | --- |
> | THD-BAR-Huge | GPT-2 | 65.3 | 68.4 |
> | THD-BAR-Huge | Qwen2.5-1.5B | 66.5 | 70.1 |
> | KAST-BAR-Large | Qwen2.5-1.5B | 70.8 | 73.9 |
>
> As shown above, upgrading THD-BAR's backbone to Qwen2.5 yields only 1.2% gain on TUEV and 1.7% on HMC, primarily due to the LLM's enhanced contextual understanding. While the stronger LLM provides a minor baseline improvement, it falls significantly short of KAST-BAR's final performance (70.8% and 73.9%). This confirms that KAST-BAR's superior performance is not primarily driven by scaling foundation LLM, but fundamentally stems from our proposed architectural innovations.
>
> **[W1] Limited novelty of core modules**
>
> We propose a novel foundation architecture that achieves joint **topological perception and knowledge-driven dynamic semantic reasoning** through our proposed DSHA, KASP and STAR.
>
> 1. **Topological perception**: Unlike unidirectional baselines (e.g., THD-BAR), which are limited in capturing both global and local dynamics, our DSHA employs a **bidirectional hierarchical interaction strategy** over the BTH, as illustrated in Figure 3. This design explicitly models EEG signals on the brain’s 3D manifold, enabling complementary coarse-to-fine and fine-to-coarse information flow and thereby preserving both macro-scale coordination and micro-scale discriminative patterns.
> 2. **Knowledge-driven Dynamic Semantic Reasoning**: Beyond static, dataset-level alignment, our **EEG refinement process is dynamic and instance-specific**. KASP generates physically grounded semantic priors for each sample, and STAR uses these priors to calibrate latent experts that adaptively aggregate and refine EEG representations through topological attention. This makes the refinement process **task-aware, instance-specific, and semantically guided**, rather than a generic text-injection or static alignment strategy.

---

> > ### Author Rebuttal · Reviewer_rP81 · 2026-04-02
> >
> > We thank the authors for their response. The authors have addressed our primary concerns regarding the paper, and we will maintain our score.

---

> > > ### Author Response · Authors · 2026-04-02
> > >
> > > Dear Reviewer rP81,
> > >
> > > Thank you for taking the time to read our rebuttal and confirming that it has addressed your primary concerns. We truly appreciate your constructive feedback, which has been invaluable in improving the clarity and quality of our manuscript.
> > >
> > > Thank you again for your support and guidance!

---

### Official Review · Reviewer_9rJx · 2026-03-11

**Soundness:** 4
**Presentation:** 3
**Significance:** 4
**Originality:** 3
**Overall Recommendation:** 4
**Confidence:** 4

**Summary:**

This paper introduces KAST-BAR, a novel foundation model framework designed to learn universal representations for cross-modal analysis by synergizing knowledge-driven semantic reasoning with dynamic topological perception. The core contribution is a three-module architecture: (1) a Dual-Stream Hierarchical Aggregator (DSHA) that encodes EEG signals by capturing both fine-grained local details and global brain topology via a bidirectional “Zoom-in/Zoom-out” mechanism; (2) a Knowledge-Anchored Semantic Profiler (KASP) that uses a frozen LLM to convert raw EEG features (temporal, spectral, spatial) into descriptive textual summaries (Semantic Profiles) without task-specific labels; and (3) a Semantic Text-Aware Refiner (STAR) that uses learnable latent queries, calibrated by the KASP text, to dynamically aggregate the most relevant discrete EEG tokens. The model is pre-trained on a large corpus of 21 datasets and fine-tuned on six diverse downstream tasks (e.g., emotion recognition, sleep staging, abnormality detection). The results demonstrate that KAST-BAR, particularly the 2.2B parameter version, achieves state-of-the-art performance compared to existing multi-task foundation models and is competitive with single-task specialists.

**Compliance With Llm Reviewing Policy:**

Affirmed.

**Final Justification:**

The authors addressed my concerns during the discussion phase with reasonable clarifications and additional ablation results, which increased my confidence for the assessment. I will maintain my original score of 4 (Weak Accept).

**Key Questions For Authors:**

1.	The bidirectional “Zoom-in/Zoom-out” interaction is a key innovation over THD-BAR. Could you provide more insight into how the information flow differs between the two streams? For example, are there specific ablation results (perhaps looking at intermediate representations) that demonstrate what unique features each stream captures?
2.	The ablation in Table 4 validates the presence of STAR, but could you provide an ablation study on the orthogonality loss itself? How does this penalty affect the diversity of the learned latent queries, and does it have a measurable impact on the model's ability to handle multiple tasks simultaneously?
3.	Could you elaborate on the computational cost and inference latency introduced by generating the Semantic Profiles with a 7B parameter LLM?

**Limitations:**

yes

**Strengths And Weaknesses:**

Strengths:
The paper presents a novel and well-motivated architecture. The combination of a dual-stream hierarchical encoder (DSHA) with a knowledge-anchored semantic profiler (KASP) and a query-based refiner (STAR) is a creative and thoughtful integration of ideas from neuroimaging, self-supervised learning, and multimodal LLMs. The technical claims are well-supported by a comprehensive experimental setup. The pre-training corpus (21 datasets) and downstream evaluations (6 tasks) are extensive and representative. The paper is well-written and structured. The narrative flows logically from problem motivation, to architectural details, to experimental validation. The figures (e.g., the overview diagram, attention visualizations) are high-quality and greatly aid understanding.

Weaknesses:
While the ablation study in Table 4 is strong, its analysis could be deepened. The specific contribution of the proposed bidirectional interaction in the DSHA encoder, compared to a simpler unidirectional propagation, could be isolated and evaluated beyond the final task results. Furthermore, the orthogonality penalty is introduced to prevent mode collapse in the STAR module, but its specific effect is not analyzed. An explicit ablation on the weight of this loss or a qualitative analysis of the learned latent queries with and without the penalty would strengthen the claim. The overall quality is high, but there are some surface-level presentation issues. Specifically, there are formatting errors such as missing backspaces (e.g., in lines 42 and 151) and missing punctuation at the end of a sentence (line 106).

---

> ### Author Rebuttal · Authors · 2026-03-31
>
> We sincerely thank Reviewer 9rJx for their positive feedback, recognizing our novel architecture (DSHA + KASP + STAR), comprehensive experimental setup, and high-quality presentation. We have carefully addressed your insightful questions below, and all updates will be incorporated into the final version.
>
> **[Q1 / W1] Information flow differences in DSHA vs. unidirectional propagation**
>
> This bidirectional design mitigates the information loss inherent in unidirectional propagation by allowing coarse-to-fine and fine-to-coarse interactions. Specifically, the two streams capture distinct feature sets:
>
> **Global Refinement Steam (Zoom-in):** Initializes at the whole-brain level and progressively queries finer scales. This top-down information flow isolates macroscopic states and cross-regional synchronization, capturing broad functional connectivity.
>
> **Local Context Steam (Zoom-out):** Initializes at individual electrodes and progressively integrates coarser scale information. This bottom-up flow preserves transient micro-events, such as focal epileptic spikes, while using broader regional consistency to filter out local artifacts.
>
> **Experimental results** further support above analysis: replacing our bidirectional DSHA with unidirectional THVQ-VAE (Yang et al., NeurIPS, 2025) removes this parallel processing capability, leading to higher reconstruction losses and a sharp decline in downstream accuracy (dropping from 70.8% to 67.4% on TUEV, as shown in Table 4). Additionally, Figure 4(a) shows replacing DSHA with THVQ-VAE leads to higher reconstruction losses during the tokenizer training phase.
>
> **[Q2 / W2] Ablation on the Orthogonality Loss $\mathcal{L}_{orth}$**
>
> The orthogonality penalty $\mathcal{L}\_{orth}$ prevents latent query redundancy, forcing experts to **capture diverse, non-overlapping semantic-topological patterns**. To evaluate its impact, we removed $\mathcal{L}\_{orth}$  from the training objective and tested the model on the TUEV and HMC datasets. To quantify **expert collapse**, we tracked the raw $\mathcal{L}_{orth}$ values during evaluation.
>
> Without the penalty, $\mathcal{L}_{orth}$ remains high (0.81), indicating that the latent experts collapse into redundant representations. Activating the penalty drops this metric to 0.03, ensuring expert diversity. This structural diversity directly improves multi-task feature aggregation, yielding accuracy gains of 2.5% on TUEV and 1.8% on HMC. We will add the table below and visual attention heatmaps comparing collapsed and diverse queries to the revised version.
>
> | Model Variant | Measured $\mathcal{L}_{orth}$ | TUEV (B-Acc) | HMC (B-Acc) |
> | --- | --- | --- | --- |
> | KAST-BAR w/o $\mathcal{L}_{orth}$ | 0.81 | 68.3 | 72.1 |
> | KAST-BAR (Full) | 0.03 | 70.8 | 73.9 |
>
> **[Q3] Computational cost and inference latency of the 7B LLM in the KASP module**
>
> **Offline Training and Evaluation:** We designed KASP as an asynchronous, offline preprocessing step for both the pre-training and standard fine-tuning/evaluation pipelines. The Semantic Profiles are generated and cached prior to training the core foundation model. Therefore, this step **does not introduce any computational** **overhead** or latency during the actual training loops or standard benchmark evaluations.
>
> **Online Real-Time Inference:** For end-to-end deployment requiring dynamic prompt generation, we benchmarked the complete KASP pipeline (feature extraction + Qwen2.5-7B inference) on a single NVIDIA L40S GPU. It takes **8 seconds per sample, which is acceptable** for clinical diagnostics. As noted in our Limitations section, future work will use knowledge distillation to transfer these 7B-generated priors into lightweight edge models to achieve millisecond-level inference.
>
> **[W3] Surface-level presentation issues**
>
> We appreciate the reviewer pointing out these typos. The missing spaces and punctuation have been corrected, and the final version will be thoroughly proofread.

---

> > ### Author Rebuttal · Reviewer_9rJx · 2026-04-02
> >
> > Thank for providing the rebuttal. The authors have addressed the concerns I raised in a constructive and thorough manner.

---

> > > ### Author Response · Authors · 2026-04-02
> > >
> > > Dear Reviewer 9rJx,
> > >
> > > Thank you for taking the time to review our rebuttal and confirming that it thoroughly addressed your concerns. Your constructive feedback was invaluable in strengthening our manuscript, and we sincerely hope these revisions give you the confidence to consider raising your score.
> > >
> > > Thank you again for your time and invaluable guidance!

---

### Official Review · Reviewer_C1ke · 2026-03-12

**Soundness:** 2
**Presentation:** 3
**Significance:** 2
**Originality:** 2
**Overall Recommendation:** 2
**Confidence:** 4

**Summary:**

This paper proposes a EEG foundation model with  1) a dual-stream hierarchical attention encoder, 2) a novel semantic-aware refiner, 3) a KASP module that employs LLM to generate dataset profiles. Experiments show state-of-the-art performance and validate the effectiveness of some proposed modules.

**Compliance With Llm Reviewing Policy:**

Affirmed.

**Final Justification:**

I still have the following 3 main concerns/questions despite the author's effort in rebuttals of all rounds.

1. The authors claim comparable task distributions to NeuroLM, but the datasets are not identical. Therefore, it remains unclear whether the performance gains arise from the proposed architecture or differences in data. The author's response is not convincing enough. Further ablation experiments for the dataset mixture are necessary.

2. Although the author reiterates the contributions in the reply, many of them are not convincing and the technical novelty is still not strong enough for a machine learning conference.

3. An analysis or visualization showing which types of text in the semantic profile are most informative for model predictions across different tasks would be recommended to demonstrate the effectiveness of the proposed module. The author said visualization figures are not allowed due to the rebuttal policy, but the policy does allow external anonymous links for such figures.

**Key Questions For Authors:**

I am confident that I understand the paper and do not have problems.

**Limitations:**

The author does not include a limitation section.

**Strengths And Weaknesses:**

Strength:
1. Some of the proposed modules, such as the dual-stream hierarchical attention and the KASP module, are interesting and potentially useful.
2. The method demonstrates notable performance improvements on several downstream tasks.

Weakness:
1. Although the paper reports performance gains, it is unclear whether these improvements stem primarily from the substantially larger model size. Most baseline models contain fewer than 5M parameters, whereas the proposed model has approximately 0.8B parameters, making the comparison questionable.
2. In addition to the difference in model size, the paper uses different mixtures of pretraining datasets, which further complicates fair comparisons with the baselines.
3. The ablation studies are limited in scope. Evaluating the proposed components on a broader range of datasets would help demonstrate the consistency and robustness of the reported improvements.
4. While some modules are interesting, the overall level of technical novelty appears limited. For instance, the EEG refinement module has already been widely adopted in prior work.
5. It would be valuable to analyze the specific contribution of KASP compared with fixed descriptions. For example, understanding what types of descriptions lead to improved performance could provide deeper insights into the mechanism behind the gains.

---

> ### Author Rebuttal · Authors · 2026-03-31
>
> We sincerely thank Reviewer C1ke for the constructive feedback and for recognizing the value of our DSHA and KASP modules, as well as the superior performance of our method across downstream tasks. Our point-by-point responses are provided below:
>
> **[W1] Performance gains from model size (0.8B/2.2B vs. <5M)**
>
> We clarify that our primary comparison targets are multi-task foundation models of **comparable or larger model sizes**, not just small-scale single-task models. As detailed in Section 4.2 (Table 2-3), leading baselines like NeuroLM-XL (Jiang et al., ICLR 2025) and THD-BAR-Huge (Yang et al., NeurIPS 2025) contain 1.7B and 1.6B parameters, respectively. In contrast, our 0.8B KAST-BAR-Base consistently outperforms these larger multi-task models across multiple datasets. This demonstrates that our performance gains stem from architectural innovations (DSHA, KASP, STAR) rather than simply scaling up parameters.
>
> | Methods | Parameters |
> | --- | --- |
> | NeuroLM-XL | 1.7B |
> | THD-BAR-Huge | 1.6B |
> | KAST-BAR-Base | **0.8B** |
> | KAST-BAR-Large | **2.2B** |
>
> **[W2] Fairness of pre-training dataset mixtures**
>
> For a fair comparison, our pre-training corpus comprises **21 standard open-source datasets**, with task distributions closely **matching baselines** like NeuroLM and THD-BAR. As shown below, our Ttotal EEG data scale is approximately 3,130 hours, which is comparable to LaBraM(\~2,500 h) and THD-BAR(\~2,100 h), while remaining far smaller than CBraMod (\~27,000 h) and NeuroLM (\~25,000 h). Nevertheless, KAST-BAR-Base still achieves consistently superior downstream performance, such as 70.8% on TUEV. This confirms that our gains are driven by the high data efficiency of our architecture (DSHA, KASP, STAR), rather than data scaling or specialized datasets.
>
> | Methods | EEG Data Scale (h) |
> | --- | --- |
> | LaBraM (Jiang et al., ICLR 2024) | ~2,500 |
> | CBraMod (Wang et al., ICLR 2025) | ~27,000 |
> | NeuroLM (Jiang et al., ICLR 2025) | ~25,000 |
> | THD-BAR (Yang et al., NeurIPS 2025) | ~2,100 |
> | KAST-BAR (Ours) | ~3,130 |
>
> **[W3] Narrow scope of ablation studies**
>
> Beyond TUEV, we further conducted ablation studies on SEED (emotion, 62-ch) and HMC (sleep, 4-ch). Replacing core component degrades performance, strictly aligning with TUEV trends. Please refer to **Reviewer kXpE [Q1/W1]** for the full cross-task table. Briefly:
>
> 1. **DSHA:** Outperforms topology-unaware VQ by **+5.6%~+9.9%** and unidirectional THVQ by **+0.3%~+3.4%**.
> 2. **STAR:** Outperforms a static backbone by **+1.4%~+4.1%**.
> 3. **KASP:** Outperforms fixed dataset-level descriptions by **+0.9%~+2.8%**.
>
> These consistent results validate our architecture's capability to extract universal representations.
>
> **[W4] Technical novelty of the overall framework and EEG refinement**
>
> We propose a novel foundation architecture that achieves joint **topological perception** and **knowledge-driven dynamic semantic reasoning** through our proposed DSHA, KASP and STAR.
>
> 1. **Topological perception**: Unlike unidirectional baselines (e.g., THD-BAR) that struggle to capture global-local dynamics, our DSHA employs a bidirectional strategy based on the Brain Topology Hierarchy (BTH), as shown in Figure 3. This explicitly models the brain's 3D manifold, perfectly balancing macro-scale coordination with micro-scale precision.
> 2. **Knowledge-driven Dynamic Semantic Reasoning:** Beyond static, dataset-level alignment, our **EEG refinement** is highly dynamic. The STAR module leverages KASP's instance-pecific priors to calibrate latent experts. These calibrated experts then directly refine EEG features by adaptively regulating topological attention, offering a fundamentally new EEG refinement module for foundation models.
>
> **[W5] Contribution of KASP vs. fixed descriptions**
>
> KASP's core advantage is dynamic, instance-specific semantic profiles from each EEG sample. Using label-free templates (Figure 9), it compels the LLM to act purely as an objective 'Data Analyst', bridging low-level physiological features and high-level medical semantics while strictly preventing hallucinations.
>
> 1. **Instance-Specific Generation (Figure 8):** KASP extracts highly instance-specific profiles. For an HMC sleep sample, it translates physical inputs into specific spatial features like *"C3 shows highest Delta power".* Conversely, for a SEED emotion sample, it explicitly highlights *"Delta and Gamma activity in F5 and F7"*.
> 2. **Performance Gains (Ablations):** These individual profiles are indispensable. Compared to static, dataset-level 'Fixed' descriptions, KASP's dynamic generation yields consistent accuracy boosts of **+0.9%~+2.8%** across heterogeneous datasets.
>
> Ultimately, as visualized in Figures 11–14, these flexible semantic priors enable the STAR module to perform instance-specific, text-guided mining of critical brain regions, which is fundamentally different from fixed descriptions that remain identical across all samples within a dataset.

---

> > ### Author Rebuttal · Reviewer_C1ke · 2026-04-04
> >
> > [W2] The authors claim comparable task distributions to NeuroLM, but the datasets are not identical. NeuroLM uses 13 datasets, whereas the proposed model uses 21, introducing potential confounding factors from dataset composition.
> >
> > Additionally, the claim of improved data efficiency based on smaller pre-training scale (vs. CBraMod and NeuroLM) is not fully convincing. Smaller datasets can outperform larger ones due to higher quality or better curation.
> >
> > Therefore, it remains unclear whether the performance gains arise from the proposed architecture or differences in data.
> >
> > [W3] Although the author reiterates the contributions, I still think the technical novelty is not significant. The STAR module is just cross-attention.
> >
> > [W5] I believe the authors may have misunderstood my comment. My request is simply for an analysis or visualization showing which types of text in the semantic profile are most informative for model predictions across different tasks.

---

> > > ### Author Response · Authors · 2026-04-05
> > >
> > > [W2] Confounding factors of dataset composition
> > >
> > > We clarify that our improvements are mainly driven by the proposed architecture, based on the following evidence.
> > >
> > > 1. **Comparison of pre-training datasets composition with baselines:** There is **no unified or identical pre-training data mixture** shared across existing EEG foundation models, including LaBraM, NeuroLM, EEGPT, THD-BAR, CBraMod, and Uni-NTFM. Following them, we constructed a comprehensive pre-training corpus consisting of 21 datasets. Importantly, we did not introduce any proprietary, private, or "specialized" datasets to favor our method. All included datasets are standard public benchmarks that have been widely adopted in prior literature. From the table below, our pre-training data heavily overlaps with the baselines across different tasks. Therefore, our performance gains do not come from special data.
> > >
> > > |Method|The same pre-training dataset as ours|
> > > |---|---|
> > > |EEGPT|SEED-IV/V/GER/FRA|
> > > |LaBraM|BCIC IV 1, Emobrain, Grasp & Lift, Inria BCI, EEGMMI, Raw/Resting EEG, SEED-IV/V/GER/FRA, Siena Scalp, SPIS, Target vs. Non-Target, TUSZ, TUAR, TUEP|
> > > |NeuroLM|BCIC IV 1, SEED-IV/V/GER/FRA, Emobrain, Grasp & Lift, Inria BCI, EEGMMI, Raw/Resting EEG, Siena Scalp, SPIS, Target vs. Non-Target, TUSZ, TUAR, TUEP|
> > > |THD-BAR|SEED-IV/V/GER/FRA, Emobrain, Grasp & Lift, Inria BCI, EEGMMI, Raw/Resting EEG, Siena Scalp, SPIS, Target vs. Non-Target, TUSZ, TUAR, TUEP|
> > > |CBraMod|TUSZ, TUAR, TUEP|
> > >
> > > 2. **Comparison of total time & Scaling Laws:** Prior works (CBraMod, EEGPT) have validated that EEG foundation models follow **scaling laws**, where model performance predictably improves as the time of pre-training data increases. Despite this, our model achieves superior performance using less length of data (e.g., ~3,130h vs. NeuroLM’s ~25,000h). This proves our gains stem from our architecture rather than massive data scaling.
> > >
> > > 3. **Decoupling Architecture from Data via Ablation:** To prove our gains come from the architecture and not the data, we trained all variants on the **exact same 21 datasets**, in ablation study. **w/o DSHA** (using NeuroLM's VQ): Accuracy drops by **5.6%~9.9%** across three tasks. **w/o KASP** (using NeuroLM's static text): Accuracy drops by **0.9%~2.8%** across three tasks. Since the data is identical, this proves our gains is driven entirely by our architecture, not the data.
> > >
> > > In summary, the above evidence suggests that the observed gains are primarily attributable to the proposed architecture, rather than to data confounders.
> > >
> > > **[W3] Novelty of the STAR module**
> > >
> > > The novelty of STAR lies in pioneering a dynamic **"Knowledge Anchoring-Guided Refinement"** paradigm. Unlike conventional multimodal models where cross-attention often acts as a generic text-injection or static, dataset-level alignment strategy, our EEG refinement process is **highly dynamic and instance-specific**. STAR leverages physically grounded **semantic priors** (generated by KASP for each sample) to actively calibrate latent experts. These calibrated experts then adaptively aggregate and refine EEG representations through topological attention, effectively extracting task-relevant fine-grained features from the DSHA-encoded brain space.
> > >
> > > Ultimately, this makes the entire refinement process task-aware, instance-specific, and semantically guided, which is an architectural paradigm shift that is unprecedented in current EEG foundation modeling.
> > >
> > > **[W5] Informative text types across downstream tasks**
> > >
> > > Thank you for the clarification. Due to the text-only constraints of the rebuttal format, we are unable to provide the requested visualization figures here. We will include the **semantic profile attention heatmaps** and conduct **detailed case studies on a broader range of samples** in the final version. Actually, visual analysis is already included in Section 4.4 (Figures 5 and 6) of our current version. In these figures, we mapped the **KASP Output Semantic Profile Keywords** (bottom panels) to the model's topological attention to demonstrate which text types drive predictions across tasks:
> > >
> > > 1. **Topological and Functional Descriptions**: As shown in the bottom panel of Figure 5, we isolate specific semantic priors such as "Cognitive Regulation" (mapped to Frontal: Fp1, Fp2...), "Emotion Processing" (Temporal: T7, T8...), and "Visual Perception" (Occipital: O1, O2...). The visualization proves that driven by these specific text types, distinct latent experts (e.g., Expert 16 bridging the Occipital and Frontal lobes) adaptively attend to these regions to construct a instance-specific network.
> > > 2. **Temporal and Spectral Features**: Figure 6 illustrates that for sleep staging, text describing spectral events like "Sleep Spindles" (Central: C3, C4) or "Slow Wave Activity" (Frontal: F4) are the most informative drivers. The heatmaps demonstrate that these specific textual priors actively redirect the latent experts to focus strictly on these precise physiological sites.

---

### Official Review · Reviewer_kXpE · 2026-03-13

**Soundness:** 3
**Presentation:** 4
**Significance:** 3
**Originality:** 3
**Overall Recommendation:** 4
**Confidence:** 3

**Summary:**

This paper proposes KAST-BAR, an EEG foundation model that combines topology-aware tokenization with LLM-based semantic guidance. The main idea is to derive sample-level semantic profiles from EEG features and use them to refine EEG representations during multimodal autoregressive pretraining. The paper evaluates the model on 6 downstream tasks after pretraining on 21 EEG datasets, and reports consistent improvements over prior multitask EEG foundation models.

**Compliance With Llm Reviewing Policy:**

Affirmed.

**Final Justification:**

The authors have addressed the ablation and generalization issues. Their reply to **Reviewer C1ke** also showed that the model size is comparable to NeuroLM-XL, but the required training data is much less.  I therefore increase to a positive score.

**Key Questions For Authors:**

1.How well do the conclusions generalize across tasks, given that the main ablation study is concentrated on a single benchmark?

2.How robust are the LLM-generated semantic profiles in KASP? Could the authors provide multiple representative examples and discuss their consistency across samples.

3.Have the authors considered using a functionally informed BTH instead of the current spatial/anatomical hierarchy?

**Limitations:**

The generated semantics are descriptive priors rather than clinically validated interpretations, and the risk of LLM hallucination remains a non-negligible limitation.

**Strengths And Weaknesses:**

**Strengths**

1.Build a more universal EEG foundation model that jointly models EEG topology and medical semantics.

2.The overall framework is well motivated and reasonably coherent, with clear functional roles for DSHA (topology-aware tokenization), KASP (knowledge-anchored semantic profiling), and STAR (semantic-guided refinement).

3.Use a relatively broad experimental setup, with pretraining on 21 EEG datasets and evaluation on 6 downstream tasks, which strengthens the empirical scope of the paper.

**Weaknesses**

1.The ablation study is narrow, with the main component analysis conducted primarily on a single downstream task.

2.The proposed dual-stream topology encoder is not compared against a sufficiently diverse set of strong topology-aware baselines, so its specific architectural advantage is not fully established.

3.The knowledge-anchoring module relies on LLM-generated semantics, but its reliability is insufficiently validated.

---

> ### Author Rebuttal · Authors · 2026-03-31
>
> We sincerely thank the reviewer for the constructive suggestions and positive assessment of our universal modeling and broad experiments.
>
> **[Q1 / W1] Narrow ablation study and generalization across tasks**
>
> Beyond TUEV, we further extended the ablation studies to two additional datasets, SEED (emotion, 62-channel) and HMC (sleep, 4-channel), to demonstrate the generalizability of our framework. The results below consistently validate the effectiveness of each core component, including DSHA, STAR, and KASP.
>
> | Tokenizer | STAR | KASP | TUEV (Event) | SEED (Emotion) | HMC (Sleep) |
> | --- | --- | --- | --- | --- | --- |
> | VQ | ✔ | ✔ | 60.9 | 68.7 | 65.2 |
> | THVQ | ✔ | ✔ | 67.4 | 74.0 | 70.6 |
> | DSHA | ✘ | ✔ | 66.7 | 72.9 | 70.1 |
> | DSHA | ✔ | ✘ | 68.0 | 73.4 | 72.5 |
> | DSHA | ✔ | ✔ | **70.8** | **74.3** | **73.9** |
>
> Substituting any core component causes performance degradation across datasets:
>
> 1. **DSHA vs. VQ & THVQ:** Replacing our bidirectional DSHA with the topology-unaware VQ or unidirectional THVQ causes accuracy drops, with DSHA achieving performance gains ranging from **+5.6% to +9.9%** (vs. VQ) and **+0.3% to +3.4%** (vs. THVQ).
> 2. **STAR vs. Static Backbone:** Downgrading the dynamic STAR refiner to a static projection reduces accuracy across all tasks, proving STAR's contribution with performance gains ranging from **+1.4% to +4.1%**.
> 3. **KASP vs. Fixed Descriptions:** Replacing KASP with fixed dataset-level text yields declines. KASP's instance-specific dynamic priors provide performance boosts of **+0.9% to +2.8%** across the three datasets.
>
> This robust cross-task consistency demonstrates the efficacy of our proposed DSHA, STAR and KASP across all three datasets.
>
> **[Q2 / W3 / Limitation] Reliability, robustness, and hallucination risk of LLM-generated semantics**
>
> To ensure KASP's reliability and mitigate LLM hallucination risks, we designed a rigorous framework grounded in strict **Prompt Constraints**. KASP uses **label-free and objectivity-oriented** prompt templates (Appendix D, Figure 9) that restrict LLM to act as a "data analyst". By forcing it to describe physical EEG signal characteristics rather than making subjective diagnostic judgments, we ensure generated semantic profiles are rooted in the input data. We validate its effectiveness through qualitative and quantitative analyses:
>
> 1. **Qualitative Validation:** Appendix Figure 8 demonstrates KASP's ability to consistently generate personalized, instance-specific profiles. For example, in the SEED sample, the profile highlights *"...Delta and Gamma activity in F5 and F7..."* alongside the prior knowledge that *"...prefrontal for regulation..."*. These precise descriptions align with the attention heatmap in Figure 5, confirming our qualitative accuracy.
> 2. **Quantitative Validation:** Replacing KASP’s dynamic generation with fixed, dataset-level text causes accuracy drops across three heterogeneous datasets (up to -2.8% on TUEV). This confirms KASP effectively extracts task-relevant, instance-specific guidance.
>
> We will open-source all generated semantic profiles, proving KASP provides stable, hallucination-free priors capturing dynamic variations.
>
> **[Q3] Functionally informed BTH vs. spatial/anatomical hierarchy**
>
> The Brain Topology Hierarchy (BTH) integrates **task-driven functional brain regions** with the **topological spatial structure** of the 10-10 system, significantly enhancing representational ability across diverse EEG datasets. For instance, Figure 7 (B2) shows the **prefrontal region** acting as a dedicated node for emotional and cognitive processing. Because of this functional-spatial alignment, this topological hierarchy is widely adopted in recent leading models like MMM (Yi et al., NeurIPS 2023) and THD-BAR (Yang et al., NeurIPS 2025) to capture cross-regional information flow.
>
> Crucially, our framework is not limited to this static hierarchy. By combining the BTH's robust cross-task representation with our STAR's **dynamic adaptability**, we extract instance-specific, task-relevant topological attention patterns  across brain regions. This dynamic regional focus is validated by the spatial attention heatmaps in Appendix Figures 11-14.
>
> **[W2] Compared with strong topology-aware baselines**
>
> We have already compared our DSHA with a **strong topology-aware baseline THVQ-VAE (Yang et al., NeurIPS 2025)**. Quantitatively, DSHA yields a +3.4% accuracy improvement over THVQ-VAE on TUEV (Table 4). Qualitatively, its training curves exhibit a steeper initial descent and a lower final reconstruction loss (Figure 4a). These results highlight the limitations of unidirectional extraction: unlike single-stream encoders that fail to integrate local dynamics with global contexts, our dual-stream DSHA explicitly models bidirectional interactions to capture the non-Euclidean manifolds of EEG. We will clarify this comparison in the final revision.

---

> > ### Author Rebuttal · Reviewer_kXpE · 2026-04-03
> >
> > I sincerely thank the authors for patiently addressing my questions! The ablation study and generalization validations are helpful. I have increased my score.

---

> > > ### Author Response · Authors · 2026-04-03
> > >
> > > Dear Reviewer kXpE,
> > >
> > > Thank you for taking the time to read our rebuttal and confirming that it thoroughly addressed your concerns. We are absolutely thrilled that the expanded ablation studies and generalization validations met your expectations, and we deeply appreciate that you increased your score.
> > >
> > > Thank you again for your time and invaluable guidance!

---

### Decision · Program_Chairs · 2026-04-30

**Decision:**

Accept (regular)

**Comment:**

This work introduces a new EEG foundation model which a) refines modeling of the spatio-temporal signal topology and b) integrates physiological representation learning with semantic LLM-based expert medical information.

The paper received disparate reviews, with three weak accepts and one clear reject. Most referees raised the point that ablation studies were insufficient, but it seems the authors have satisfactorily addressed this in their rebuttal. Two referees (C1ke & rP81) questioned the novelty of the approach, while one referee (9rJx), in contrast, found the approach creative and thoughtful. After my own reading of the paper, I tend to agree with the latter - I think the paper *does* provide a quite novel way of, in particular, integrating LLM knowledge with the foundational representation of the EEG data. One referee (C1ke) argued the comparisons in the paper were not fair, with the present model much larger in parameter size than baselines tested against, adding to differences among models in datasets used for pretraining. However, I find the authors' rebuttal to this point reasonably convincing, as other EEG foundation models of comparable size had indeed been included.

The additional ablation experiments the authors provided during the rebuttal I find particularly important, and I'd also appreciate a bit of a deeper dive in the revision into how, why, when, and in which aspects the textual descriptions help improving performance.